# Form and Volume of the Stem of *Tectona grandis* L.f. in the Central-WESTERN Region of Brazil

**Karen Janones da Rocha** [1,*], **César Augusto Guimarães Finger** [2], **Cyro Matheus Cometti Favalessa** [3], **Sidney Fernando Caldeira** [3] and **Frederico Dimas Fleig** [2]

1 Laboratory of Ecosystem Recovery and Forest Production, Academic Department of Forest Engineering, Federal University of Rondônia, Rolim de Moura 76940-000, Brazil
2 Postgraduate Program in Forest Engineering, Federal University of Santa Maria, Santa Maria 97105-900, Brazil
3 Faculty of Forestry, Federal University of Mato Grosso, Cuiabá 78060-719, Brazil
* Correspondence: karenrocha@unir.br

**Abstract:** The international market has recognized the high value of *Tectona grandis* L.f. plantations, requiring the development of reliable and accurate tools and techniques to quantify forest stocks accurately. In this study, we developed suitable equations to estimate the stem diameters and volume of *Tectona grandis* trees in the central-western region of Brazil, evaluating the stem form change points (FCPs) and testing the stratification of data as a measure to control their variation. The Schöepfer model was tested in the study of the FCPs of the stem, and single equation, segmented and variable-exponent taper functions were used to describe the stem profile. After the selection of the model for the taper, data stratification in the DBH classes, form parameter "r" and artificial form factor were proposed. The total volumes of each tree were calculated by integrating the Clark III et al. model. The FCPs of the *Tectona grandis* stems occurred at 28%, 57% and 73% of the total height, corresponding, on average, to the absolute positions of 6.4 m, 13.3 m and 16.5 m. The Clark III et al. equation, without stratification, was the most appropriate equation to estimate the diameters along the stem and the volume of *Tectona grandis* trees in the central-western region of Brazil.

**Keywords:** taper; stratification; teak

## 1. Introduction

The strong demand for *Tectona grandis* L.f. wood from the family Lamiaceae [1], popularly known as teak, in the international market makes its plantations important sources of income and investments for producers, besides offering advantages over other tropical and temperate species due to their growth characteristics and quality of their wood. Teak wood demonstrates a combination of beauty, stability, durability, and resistance; is widely used in shipbuilding; and is suitable for luxury outdoor and indoor furniture due to its weathering resistance, seasoning without splitting and cracking, lightness and strength [2].

In recent decades, the international market seems to have recognized a higher value for teak plantations, whose log prices have been rising rapidly [3]. Therefore, the worldwide acreage of teak plantations has increased over recent decades. Many scientific studies were published about different aspects of teak growth and yield from different parts of the world, including Brazil, China, Colombia, Costa Rica, Ecuador, Ghana, Guatemala, India, Nigeria and Panama [4–14]. Brazil is the country with the largest planted area in Latin America with 93,957 ha, which is distributed in the states of the northern and central-west regions [15].

The increasingly competitive timber market requires the development and application of techniques and tools to reduce the costs and waste in the production processes in the forestry undertaking. Thus, quantifying forest stocks accurately requires recurrent studies related to the volume of trees, either individually or throughout the stands. In forest

production, especially sawn wood, the objective is to maximize it by reducing waste in wood transformation, whether primary or secondary. Thus, research related to volume modeling and stem tapering is fundamental to gathering reliable and precise information about forest products [16]. To achieve this goal, it is often necessary to use sophisticated modeling techniques, such as taper equations [17].

There are relatively fewer articles on taper modeling of teak trees, but some studies are worth mentioning, such as those of Choochuen et al. [18], Favalessa et al. [19], Koirala et al. [20], Lanssanova et al. [21], Pelissari et al. [22], Tavares Junior et al. [23] and Wijenayake et al. [24]. The taper model to be selected must be flexible enough to express the effect of the environment and forestry practices that influence the distribution of the annual increase in wood along the stem that influences the total volume, along with the quality and quantity of wood from stands. In addition, it should be checked how these functions fit the different portions and shapes of the stem.

Teak trees generally have a large buttress and they taper rapidly as the height increases; therefore, traditional volume equations that are based only on the total tree height and breast height diameter (dbh) might not address the volume loss incurred due to the rapid tapering [9,14]. This variation in diameter at any height can be modeled with the use of a taper equation. Basically, these equations estimate the diameter around the bark of the tree stem at any given height up the stem [20].

Most of the literature on taper modeling can be grouped into three types of models: single equation models, segmented models and variable-exponent models. In single equation taper models, the taper of the stem is described using a single mathematical function. One of the most popular taper models developed by Kozak et al. [25] falls into this category. However, the lack of flexibility could easily result in highly biased estimates [26].

The second group is a more complex segmented form of taper equations, where different equations are used for various parts of the stem and joined in such a way that their first derivatives are equal at the points of intersection [27]. The first was proposed by Max and Burkhart [28], and Parresol et al. [29] and Clark III et al. [30] can be cited. The segmented model proposed by Clark III et al. [30] is a combination of the best attributes of Schlaegel's and Max and Burkhart's models, managing to explain differences in tapering by form classes. This type of taper equation models a tree stem as three or more segments: a neiloid, paraboloid, cylinder and conoid that are joined to form the taper equation. In variable-exponent models, such as Kozak's [31], a single continuous function with an exponent changing from the stump to the top describes several intermediate forms. Such a power function eliminates the necessity of developing segmented taper functions for different portions of the stem in order to reduce the local bias [32].

The segmented polynomial models with estimated joining points provided an improved description of tree tapering when compared with a single taper model that is used throughout the stem length [28]. These joining points, which are transition points from one solid shape to another, are usually as follows: (1) below 15%—change of shape from a neoloid to a paraboloid, and (2) above 75%—change of shape from a paraboloid to a cone [26,33,34].

The objective of this work was to describe the stem form with the form change points (FCPs) approach and the stratification of data aiming to control its variation, along with modifying the equation structure to estimate diameters along the stem and the total volume of trees of *Tectona grandis* L.f. in the central-western region of Brazil.

## 2. Materials and Methods

### 2.1. Study Area and Data

The study was carried out with data from four forest plantations of *Tectona grandis* L. f. in four locations in the state of Mato Grosso, Brazil (Table 1). The trees were randomly obtained in the final harvest of the stands. The climate of the region is Aw, i.e., tropical with a defined dry season and with an average temperature variation between 22 and 24 °C [35].

**Table 1.** Information on the stands of *Tectona grandis* L.f. used to study the stem forms, central-west region, Brazil.

| Farm | Coordinates | | Total Area | Useful Area | $t_0$ | $t$ | $n$ | $\bar{d}$ | $\bar{h}$ | $h_{100}$ |
|------|-----|-----|------------|-------------|-------|-----|-----|-----|-----|-----------|
| | **S** | **W** | **(ha)** | **(m² tree⁻¹)** | | **(Years)** | **(Tree)** | **(cm)** | **(m)** | **(m)** |
| Aguaçu | 15°37′ | 55°33′ | 118.5 | 6 | 2001 | 12 | 52 | 25.02 (±5.16) | 19.0 (±1.7) | 21.3 |
| Soroteca | 15°37′ | 58°18′ | 60.0 | 9 | 2000 | 8 | 63 | 18.24 (±2.48) | 17.0 (±1.3) | 17.5 |
| Teca do Brasil | 16°12′ | 56°22′ | 1260.0 | 9 | 1999–2003 | 9 to 16 | 224 | 23.04 (±8.64) | 18.7 (±3.5) | 17.9 |
| Teca do Jauru | 15°25′ | 58°37′ | 6.8 | 3 | 1979 | 36 | 31 | 40.55 (±5.16) | 24.3 (±3.2) | 27.6 |

UA—sampling unit; $t_0$—year of implantation; $t$—age of trees when scaling was performed, in years; $n$—number of trees scaled per farm; $\bar{d}$—mean diameter at breast height, in cm; $\bar{h}$—mean height, in m; $h_{100}$—dominant height of Assmann, in m.

The scaling was performed at different measuring height points ranging from the ground to the top of the tree. The distance between these points varied from one to three meters, but for all trees, the values were measured at the 1.3 m height position and at least two measurements below it. Differences in the distance between these measuring height points do not affect the accuracy of the models, as was already observed by Kozak [36] for sections up to three meters before or after the curve inflection points described by the models.

For all the trees ($j$), the diameters were measured at positions $i$ ($d_{ij}$) at various heights ranging from the ground to the top of the tree ($h_{ij}$), the diameter at 1.30 m at breast height ($d_j$) and height ($h_j$). The actual volume, both total and partial, in each portion, was obtained by adding the partial volumes of the sections with the Smalian procedure. The dominant height ($h_{100}$) was calculated by considering the concept of Assmann [37], which uses the average height of the 100 largest-diameter trees per hectare.

The scaled trees were distributed in ten diameter classes at breast height (DBH) and were randomly selected to compose the adjustment database for the taper models (Table 2). In the calculation of the ideal sampling intensity by diameter class, the population was considered infinite, with a maximum allowable error limit of 10% on average, with a 95% confidence probability, as recommended by Péllico Netto and Brena [38].

**Table 2.** Ideal sampling intensity and absolute frequency of *Tectona grandis* L.f. trees in different diameter classes, central-west region, Brazil.

| Class | Diameter Class ($d$—cm) | | | $n$ | $FA$ | $CV$ |
|-------|-------------|------------------|-------------|-----|------|------|
| | **Lower Limit** | **Center of Class** | **Upper Limit** | **(Trees)** | **(Trees)** | **(%)** |
| 1 | 10.0 | ⊢ | 15.0 | 33 | 20 | 28 |
| 2 | 15.0 | ⊢ | 20.0 | 3 | 20 | 9 |
| 3 | 20.0 | ⊢ | 25.0 | 9 | 20 | 14 |
| 4 | 25.0 | ⊢ | 30.0 | 2 | 20 | 8 |
| 5 | 30.0 | ⊢ | 35.0 | 10 | 20 | 15 |
| 6 | 35.0 | ⊢ | 40.0 | 12 | 16 | 16 |
| 7 | 40.0 | ⊢ | 45.0 | 9 | 18 | 14 |
| 8 | 45.0 | ⊢ | 50.0 | 5 | 9 | 9 |
| 9 | 50.0 | ⊢ | 55.0 | 4 | 2 | 2 |
| 10 | 55.0 | ⊢ | 60.0 | 27 | 3 | 12 |
| | **SUM** | | | | 148 | |

$n$—optimal tree sampling intensity; $FA$—absolute frequency of trees; $CV$—coefficient of variation of the variable volume in percentage.

Even when not meeting the sample intensity in all the diametric classes, according to Kozak [36], the estimates of the taper equations' parameters are simple and can be estimated with any multiple regression program with about 60 to 100 trees. Nevertheless, twenty trees in each class were selected to establish a rectangular distribution of frequencies in the DBH classes. This procedure is justified by the coincidence of the mean and center of each class, which implies the disappearance of the rounding error [39].

### 2.2. Study of the form Change Points of the Stems

The stems of trees present very varied shapes depending on several factors, and they can be perfectly similar to a defined geometric shape or difficult to define. The point where the change occurs in the form of this geometric figure is where the curve describing the profile of the stem changes in concavity, that is, the inflection point of the curve, which is mathematically obtained from the second derivative of the zero-equal function. In the case of segmented polynomial models, each segment is considered to describe a form; therefore, when the derivatives are applied in the function, the beginning/end of each segment may coincide with the points of maximum and minimum curves.

Unsegmented polynomial models of the third degree have one change point (FCP) in the stem, polynomials of the fourth grade have two FCPs, polynomials of the fifth grade have three FCPs and so on. The Schöepfer model [40], which is characterized by the unsegmented polynomial of the fifth degree, was adjusted for each of the trees used in the study of FCPs in the stem. The equations described presented all the significant coefficients, with the statistics $R^2_{aj.}$ above 0.90 and $S_{yx}\%$ below 10%. Wolfram Alpha LLC © [41] was used to calculate the first and second derivatives from the Schöepfer function [40], and from these, the FCPs were known for each tree selected. Thus, charts with the medium profile and derivatives from the function, together with the FCPs, were elaborated for the study in the form change points in the stems.

### 2.3. Models Evaluated

The single equations (Schöepfer [40], Kozak et al. (1969) [25], Demaerschalk [42], Demaerschalk [43] and Ormerod [44]), segmented equations (Max and Burkhart [28], Cao et al. [45] rewritten by Byrne and Reed [46], Parresol et al. [29], Clark III et al. [30] and Kozak (1988) [36]) and segmented variable-exponent equations (Bi [32], Lee et al. [47] and Kozak (2004) [31]) were fitted with the dependent variable diameter that was measured along the stem ($d_{ij}$), as proposed by Môra et al. [48]. Thus, the estimates of the variable $d_{ij}$ were more accurate than those obtained with the model fitted in the original form [($d_{ij}/d_j$) or ($d_{ij}/d_j$)$^2$]. In the adjustment of the Clark III et al. model [30], a diameter measurement with bark (cm) at a height of 5.3 m ($d_{5.3j}$) was necessary. For the trees that did not have a measurement at 5.3 m, linear interpolation was used to estimate these diameters.

### 2.4. Date Stratification and Change in the Model Variables

After selecting the model to describe the taper, stratification by DBH class, by form parameter class "r" and by artificial form factor class ($f_{1.3}$) of the sample trees were performed (Table 3) for the purpose of obtaining more accurate estimates of the diameters [49,50]. The diameter classes, with a range of 5 cm, were constructed from inclusion ranging from the smallest up to the largest diameter observed. The form classes were defined from the calculated values for the parameter "r" and *values* of $f_{1.3}$ calculated with the tree volume ratio and the cylinder volume with a diameter corresponding to the DBH and tree height.

The form parameter "r" was expressed using the function $g_w = p \times w^r$, where $g_w$ is the basal area of the stem at a distance $w$ from the crown edge and $p$ is the parameter that describes the size of the rotation body, where $p = a \times \pi$ and $a$ is the curve inclination. According to the variation in the exponent of form "r", different solids of revolution were generated. When "r" is 0, the form resembles the cylinder; "r" equal to 1 resembles a quadratic parabola; "r" equal to 2 resembles a cone; and when "r" is equal to 3, it describes the neiloid form [51]. All classifications cover the data amplitude and, for the parameter

variables "$r$" and $f_{1.3}$, three level classes were defined to maintain the minimum frequency of ten trees per class.

Considering the stratification result, due to the non-significance of coefficients, two approaches were proposed to the Clark III et al. model [30]. In the first approach, it was proposed that the component $\beta_3/d_j^3$ be withdrawn from the model, and for the second modification, component $\beta_2 + \beta_3/d_j^3$ was to be withdrawn (Table 4). Other authors also verified the non-significance of the coefficient $\beta_3$ and suggested similar modifications [52].

Other modifications were proposed with the study of the FCPs that involved exchanging the diameter at 5.3 m height ($d_{5.3j}$) with the diameters at 6.4 m height ($d_{6.4j}$) and 28% of the total height ($d_{0.28h}$), adding a new coefficient ($\beta_6$) with a diameter of 73% of the total height ($d_{0.73h}$) in modifications 3, 4, 5 and 6, respectively. The use of the height-over-DBH ratio as a description of the stem form was also tested, resulting in modifications 7 and 8, which used the inverse of the height–diameter ratio ($1/hd_j$ or $d_j h$). Newnham [53] and Kozak [36] also included the d/h ratio in the functions in an attempt to describe the variation in the stem form caused by variations in the relative length of the crown.

**Table 3.** Stratification proposed by the diameter class at breast height, by parameter class "$r$" and by artificial form factor class for *Tectona grandis* L.f. scaled trees, central-west region, Brazil.

| Class | Diameter Class at Breast Height ($d$—cm) | | | Absolute Frequency |
| --- | --- | --- | --- | --- |
| | Lower Limit | Center of Class | Upper Limit | (Trees) |
| 1 | 10.0 | ⊢ | 15.0 | 20 |
| 2 | 15.0 | ⊢ | 20.0 | 20 |
| 3 | 20.0 | ⊢ | 25.0 | 20 |
| 4 | 25.0 | ⊢ | 30.0 | 20 |
| 5 | 30.0 | ⊢ | 35.0 | 20 |
| 6 | 35.0 | ⊢ | 40.0 | 16 |
| 7 | 40.0 | ⊢ | 45.0 | 18 |
| 8 | 45.0 | ⊢ | 50.0 | 9 |
| 9 | 50.0 | ⊢ | 55.0 | 2 |
| 10 | 55.0 | ⊢ | 60.0 | 3 |
| Class | Parameter "$r$" | | | Absolute Frequency |
| | Lower Limit | Center of Class | Upper Limit | (trees) |
| 1 | 0.30 | ⊢ | 0.80 | 85 |
| 2 | 0.80 | ⊢ | 1.30 | 45 |
| 3 | 1.30 | ⊢ | 1.80 | 18 |
| Class | Artificial Form Factor ($f_{1.3}$) | | | Absolute Frequency |
| | Lower Limit | Center of Class | Upper Limit | (trees) |
| 1 | 0.30 | ⊢ | 0.38 | 32 |
| 2 | 0.38 | ⊢ | 0.46 | 100 |
| 3 | 0.46 | ⊢ | 0.55 | 16 |

**Table 4.** Modifications proposed to the Clark III et al. model for the adjustment of data from *Tectona grandis* L.f., central-west region, Brazil.

| | |
|---|---|
| **Clark III et al. [30]** | $$d_{ij} = \left\{ \begin{aligned} &IS * d_j^2 \left\{ 1 + \left(\beta_2 + \frac{\beta_3}{d_j^3}\right) * \left[\frac{\left(1 - \frac{h_{ij}}{h_j}\right)^{\beta_0} - \left(1 - \frac{1.3}{h_j}\right)^{\beta_0}}{1 - \left(1 - \frac{1.3}{h_j}\right)^{\beta_0}}\right] \right\} + \\ &IB * \left\{ d_j^2 - (d_j^2 - d_{5.3j}^2) * \left[\frac{\left(1 - \frac{1.3}{h_j}\right)^{\beta_1} - \left(1 - \frac{h_{ij}}{h_j}\right)^{\beta_1}}{\left(1 - \frac{1.3}{h_j}\right)^{\beta_1} - \left(1 - \frac{5.3}{h_j}\right)^{\beta_1}}\right] \right\} + \\ &IT * \left\{ d_{5.3j}^2 * \left[\begin{aligned} &\beta_5 * \left(\frac{h_{ij} - 5.3}{h_j - 5.3} - 1\right)^2 \\ &+IM * \frac{1 - \beta_5}{\beta_4^2} * \left(\beta_4 - \frac{h_{ij} - 5.3}{h_j - 5.3}\right)^2 \end{aligned}\right] \right\} \end{aligned} \right\}^{0.5} + \varepsilon_{ij}$$ |
| 1 | $$d_{ij} = \left\{ \begin{aligned} &IS * d_j^2 \left\{ 1 + (\beta_2) * \left[\frac{\left(1 - \frac{h_{ij}}{h_j}\right)^{\beta_0} - \left(1 - \frac{1.3}{h_j}\right)^{\beta_0}}{1 - \left(1 - \frac{1.3}{h_j}\right)^{\beta_0}}\right] \right\} + \\ &IB * \left\{ d_j^2 - (d_j^2 - d_{5.3j}^2) * \left[\frac{\left(1 - \frac{1,3}{h_j}\right)^{\beta_1} - \left(1 - \frac{h_{ij}}{h_j}\right)^{\beta_1}}{\left(1 - \frac{1.3}{h_j}\right)^{\beta_1} - \left(1 - \frac{5.3}{h_j}\right)^{\beta_1}}\right] \right\} + \\ &IT * \left\{ d_{5.3j}^2 * \left[\begin{aligned} &\beta_5 * \left(\frac{h_{ij} - 5.3}{h_j - 5.3} - 1\right)^2 \\ &+IM * \frac{1 - \beta_5}{\beta_4^2} * \left(\beta_4 - \frac{h_{ij} - 5.3}{h_j - 5.3}\right)^2 \end{aligned}\right] \right\} \end{aligned} \right\}^{0.5} + \varepsilon_{ij}$$ |
| 2 | $$d_{ij} = \left\{ \begin{aligned} &IS * d_j^2 \left\{ 1 * \left[\frac{\left(1 - \frac{h_{ij}}{h_j}\right)^{\beta_0} - \left(1 - \frac{1.3}{h_j}\right)^{\beta_0}}{1 - \left(1 - \frac{1.3}{h_j}\right)^{\beta_0}}\right] \right\} + \\ &IB * \left\{ d_j^2 - (d_j^2 - d_{5.3j}^2) * \left[\frac{\left(1 - \frac{1.3}{h_j}\right)^{\beta_1} - \left(1 - \frac{h_{ij}}{h_j}\right)^{\beta_1}}{\left(1 - \frac{1.3}{h_j}\right)^{\beta_1} - \left(1 - \frac{5.3}{h_j}\right)^{\beta_1}}\right] \right\} + \\ &IT * \left\{ d_{5.3j}^2 * \left[\begin{aligned} &\beta_5 * \left(\frac{h_{ij} - 5.3}{h_j - 5.3} - 1\right)^2 \\ &+IM * \frac{1 - \beta_5}{\beta_4^2} * \left(\beta_4 - \frac{h_{ij} - 5.3}{h_j - 5.3}\right)^2 \end{aligned}\right] \right\} \end{aligned} \right\}^{0.5} + \varepsilon_{ij}$$ |
| 3 | $$d_{ij} = \left\{ \begin{aligned} &IS * d_j^2 \left\{ 1 + \left(\beta_2 + \frac{\beta_3}{d_j^3}\right) * \left[\frac{\left(1 - \frac{h_{ij}}{h_j}\right)^{\beta_0} - \left(1 - \frac{1.3}{h_j}\right)^{\beta_0}}{1 - \left(1 - \frac{1.3}{h_j}\right)^{\beta_0}}\right] \right\} + \\ &IB * \left\{ d_j^2 - (d_j^2 - d_{6.4j}^2) * \left[\frac{\left(1 - \frac{1.3}{h_j}\right)^{\beta_1} - \left(1 - \frac{h_{ij}}{h_j}\right)^{\beta_1}}{\left(1 - \frac{1.3}{h_j}\right)^{\beta_1} - \left(1 - \frac{6.4}{h_j}\right)^{\beta_1}}\right] \right\} + \\ &IT * \left\{ d_{6.4j}^2 * \left[\begin{aligned} &\beta_5 * \left(\frac{h_{ij} - 6.4}{h_j - 6.4} - 1\right)^2 \\ &+IM * \frac{1 - \beta_5}{\beta_4^2} * \left(\beta_4 - \frac{h_{ij} - 6.4}{h_j - 6.4}\right)^2 \end{aligned}\right] \right\} \end{aligned} \right\}^{0.5} + \varepsilon_{ij}$$ |

| | |
|---|---|
| **Clark III et al. [30]** | $$d_{ij} = \left\{ \begin{array}{l} IS*d_j^2\left\{1+\left(\beta_2+\dfrac{\beta_3}{d_j^3}\right)*\left[\dfrac{\left(1-\dfrac{h_{ij}}{h_j}\right)^{\beta_0}-\left(1-\dfrac{1.3}{h_j}\right)^{\beta_0}}{1-\left(1-\dfrac{1.3}{h_j}\right)^{\beta_0}}\right]\right\}+ \\[2em] IB*\left\{d_j^2-(d_j^2-d_{5.3j}^2)*\left[\dfrac{\left(1-\dfrac{1.3}{h_j}\right)^{\beta_1}-\left(1-\dfrac{h_{ij}}{h_j}\right)^{\beta_1}}{\left(1-\dfrac{1.3}{h_j}\right)^{\beta_1}-\left(1-\dfrac{5.3}{h_j}\right)^{\beta_1}}\right]\right\}+ \\[2em] IT*\left\{d_{5.3j}^2*\left[\begin{array}{l}\beta_5*\left(\dfrac{h_{ij}-5.3}{h_j-5.3}-1\right)^2 \\ +IM*\dfrac{1-\beta_5}{\beta_4^2}*\left(\beta_4-\dfrac{h_{ij}-5.3}{h_j-5.3}\right)^2\end{array}\right]\right\} \end{array}\right\}^{0.5}+\varepsilon_{ij}$$ |
| 4 | $$d_{ij} = \left\{ \begin{array}{l} IS*d_j^2\left\{1+\left(\beta_2+\dfrac{\beta_3}{d_j^3}\right)*\left[\dfrac{\left(1-\dfrac{h_{ij}}{h_j}\right)^{\beta_0}-\left(1-\dfrac{1.3}{h_j}\right)^{\beta_0}}{1-\left(1-\dfrac{1.3}{h_j}\right)^{\beta_0}}\right]\right\}+ \\[2em] IB*\left\{d_j^2-(d_j^2-d_{0.28hj}^2)*\left[\dfrac{\left(1-\dfrac{1.3}{h_j}\right)^{\beta_1}-\left(1-\dfrac{h_{ij}}{h_j}\right)^{\beta_1}}{\left(1-\dfrac{1.3}{h_j}\right)^{\beta_1}-\left(1-\dfrac{h_{0.28hj}}{h_j}\right)^{\beta_1}}\right]\right\}+ \\[2em] IT*\left\{d_{0.28hj}^2*\left[\begin{array}{l}\beta_5*\left(\dfrac{h_{ij}-h_{0.28hj}}{h_j-h_{0.28hj}}-1\right)^2 \\ +IM*\dfrac{1-\beta_5}{\beta_4^2}*\left(\beta_4-\dfrac{h_{ij}-h_{0.28hj}}{h_j-h_{0.28hj}}\right)^2\end{array}\right]\right\} \end{array}\right\}^{0.5}+\varepsilon_{ij}$$ |
| 5 | $$d_{ij} = \left\{ \begin{array}{l} IS*d_j^2\left\{1+\left(\beta_2+\dfrac{\beta_3}{d_j^3}\right)*\left[\dfrac{\left(1-\dfrac{h_{ij}}{h_j}\right)^{\beta_0}-\left(1-\dfrac{1.3}{h_j}\right)^{\beta_0}}{1-\left(1-\dfrac{1.3}{h_j}\right)^{\beta_0}}\right]\right\}+ \\[2em] IB*\left\{d_j^2-(d_j^2-d_{5.3j}^2)*\left[\dfrac{\left(1-\dfrac{1.3}{h_j}\right)^{\beta_1}-\left(1-\dfrac{h_{ij}}{h_j}\right)^{\beta_1}}{\left(1-\dfrac{1.3}{h_j}\right)^{\beta_1}-\left(1-\dfrac{5.3}{h_j}\right)^{\beta_1}}\right]\right\}+ \\[2em] IT*\left\{d_{5.3j}^2*\left[\begin{array}{l}\beta_5*\left(\dfrac{h_{ij}-5.3}{h_j-5.3}-1\right)^2+\beta_6*d_{0.73hj} \\ +IM*\dfrac{1-\beta_5}{\beta_4^2}*\left(\beta_4-\dfrac{h_{ij}-5.3}{h_j-5.3}\right)^2\end{array}\right]\right\} \end{array}\right\}^{0.5}+\varepsilon_{ij}$$ |
| 6 | $$d_{ij} = \left\{ \begin{array}{l} IS*d_j^2\left\{1+\left(\beta_2+\dfrac{\beta_3}{d_j^3}\right)*\left[\dfrac{\left(1-\dfrac{h_{ij}}{h_j}\right)^{\beta_0}-\left(1-\dfrac{1.3}{h_j}\right)^{\beta_0}}{1-\left(1-\dfrac{1.3}{h_j}\right)^{\beta_0}}\right]\right\}+ \\[2em] IB*\left\{d_j^2-(d_j^2-d_{0.28hj}^2)*\left[\dfrac{\left(1-\dfrac{1.3}{h_j}\right)^{\beta_1}-\left(1-\dfrac{h_{ij}}{h_j}\right)^{\beta_1}}{\left(1-\dfrac{1.3}{h_j}\right)^{\beta_1}-\left(1-\dfrac{h_{0.28hj}}{h_j}\right)^{\beta_1}}\right]\right\}+ \\[2em] IT*\left\{d_{0.28hj}^2*\left[\begin{array}{l}\beta_5*\left(\dfrac{h_{ij}-h_{0.28hj}}{h_j-h_{0.28hj}}-1\right)^2+\beta_6*d_{0.73hj} \\ +IM*\dfrac{1-\beta_5}{\beta_4^2}*\left(\beta_4-\dfrac{h_{ij}-h_{0.28hj}}{h_j-h_{0.28hj}}\right)^2\end{array}\right]\right\} \end{array}\right\}^{0.5}+\varepsilon_{ij}$$ |

**Table 4.** *Cont.*

| | |
|---|---|
| **Clark III et al. [30]** | $$d_{ij} = \left\{ \begin{array}{l} IS * d_j^2 \left\{ 1 + \left( \beta_2 + \dfrac{\beta_3}{d_j^3} \right) * \left[ \dfrac{\left(1 - \frac{h_{ij}}{h_j}\right)^{\beta_0} - \left(1 - \frac{1.3}{h_j}\right)^{\beta_0}}{1 - \left(1 - \frac{1.3}{h_j}\right)^{\beta_0}} \right] \right\} + \\ IB * \left\{ d_j^2 - (d_j^2 - d_{5.3j}^2) * \left[ \dfrac{\left(1 - \frac{1.3}{h_j}\right)^{\beta_1} - \left(1 - \frac{h_{ij}}{h_j}\right)^{\beta_1}}{\left(1 - \frac{1.3}{h_j}\right)^{\beta_1} - \left(1 - \frac{5.3}{h_j}\right)^{\beta_1}} \right] \right\} + \\ IT * \left\{ d_{5.3j}^2 * \left[ \begin{array}{l} \beta_5 * \left( \frac{h_{ij} - 5.3}{h_j - 5.3} - 1 \right)^2 \\ + IM * \frac{1 - \beta_5}{\beta_4^2} * \left( \beta_4 - \frac{h_{ij} - 5.3}{h_j - 5.3} \right)^2 \end{array} \right] \right\} \end{array} \right\}^{0.5} + \varepsilon_{ij}$$ |
| 7 | $$d_{ij} = \left\{ \begin{array}{l} IS * d_j^2 \left\{ 1 + \left( \beta_2 + \dfrac{\beta_3}{d_j^3} \right) * \left[ \dfrac{\left(1 - \frac{h_{ij}}{h_j}\right)^{\beta_0} - \left(1 - \frac{1.3}{h_j}\right)^{\beta_0}}{1 - \left(1 - \frac{1.3}{h_j}\right)^{\beta_0}} \right] \right\} + \\ IB * \left\{ d_j^2 - (d_j^2 - d_{5.3j}^2) * \left[ \dfrac{\left(1 - \frac{1.3}{h_j}\right)^{\beta_1} - \left(1 - \frac{h_{ij}}{h_j}\right)^{\beta_1}}{\left(1 - \frac{1.3}{h_j}\right)^{\beta_1} - \left(1 - \frac{5.3}{h_j}\right)^{\beta_1}} \right] \right\} + \\ IT * \left\{ d_{5.3j}^2 * \left[ \begin{array}{l} \beta_5 * \left( \frac{h_{ij} - 5.3}{h_j - 5.3} - 1 \right)^2 + \beta_6 * \frac{1}{hd_j} \\ + IM * \frac{1 - \beta_5}{\beta_4^2} * \left( \beta_4 - \frac{h_{ij} - 5.3}{h_j - 5.3} \right)^2 \end{array} \right] \right\} \end{array} \right\}^{0.5} + \varepsilon_{ij}$$ |
| 8 | $$d_{ij} = \left\{ \begin{array}{l} IS * d_j^2 \left\{ 1 + \left( \beta_2 + \dfrac{\beta_3}{d_j^3} \right) * \left[ \dfrac{\left(1 - \frac{h_{ij}}{h_j}\right)^{\beta_0} - \left(1 - \frac{1.3}{h_j}\right)^{\beta_0}}{1 - \left(1 - \frac{1.3}{h_j}\right)^{\beta_0}} \right] \right\} + \\ IB * \left\{ d_j^2 - (d_j^2 - d_{0.28hj}^2) * \left[ \dfrac{\left(1 - \frac{1.3}{h_j}\right)^{\beta_1} - \left(1 - \frac{h_{ij}}{h_j}\right)^{\beta_1}}{\left(1 - \frac{1.3}{h_j}\right)^{\beta_1} - \left(1 - \frac{h_{0.28hj}}{h_j}\right)^{\beta_1}} \right] \right\} + \\ IT * \left\{ d_{0.28hj}^2 * \left[ \begin{array}{l} \beta_5 * \left( \frac{h_{ij} - h_{0.28hj}}{h_j - h_{0.28hj}} - 1 \right)^2 + \beta_6 * \frac{1}{hd_j} \\ + IM * \frac{1 - \beta_5}{\beta_4^2} * \left( \beta_4 - \frac{h_{ij} - h_{0.28hj}}{h_j - h_{0.28hj}} \right)^2 \end{array} \right] \right\} \end{array} \right\}^{0.5} + \varepsilon_{ij}$$ |

$d_j$—diameter at breast height with bark (cm) measured in each tree $j$; $h_j$—height (m) in each tree $j$; $h_{ij}$—height (m) of the base of the stem to the point where the diameter $d_{ij}$ is reached in each tree $j$; $d_{ij}$—diameter (cm) with bark at a height $h_{ij}$ in each tree $j$; $d_{5.3j}$—diameter with bark measured (cm) at a height of 5.3 m in each tree $j$; $d_{6.4j}$—diameter with bark measured (cm) at a height of 6.4 m in each tree $j$; $d_{0.28h}$—diameter at 28% of the height (m); $d_{0.73h}$—diameter at 73% height (m); $h_{0.28h}$—28% height (m); $h_{0.73h}$—73% height (m); $IS$, $IB$, $IT$ and $IM$—connection points of the Clark III et al. model [30]: $IS = 1$ if $d_{ij}^2 \geq d_j^2$, otherwise $IS = 0$; $IB = 1$ if $d_j^2 > d_{ij}^2 \geq F^2$, otherwise $IB = 0$; $IT = 1$ if $F^2 > d_j^2$, otherwise $IT = 0$; $IM = 1$ if $d_j^2 > \beta_5(\beta_4 - 1)^2 F^2$, otherwise $IM = 0$; $F = d_{5.3j}^2$. For modifications 1, 2, 5 and 7: $F = d_{5.3j}^2$; for modification 3: $F = d_{6.4j}^2$; for modifications 4, 6 and 8: $F = d_{0.28hj}^2$.

### 2.5. Statistical Evaluation of the Models

All data processing was carried out using the SAS statistical system, version 9.3 [54], using the proc nlin procedures and the Marquardt algorithm to minimize the residual sum of squares. Student's *t*-test was used to evaluate the significance of the equation coefficients by testing whether the null hypothesis of the coefficient was zero at 5% significance. When only one of the coefficients of the equations obtained a non-significant result, the modification of the model that involved removing this coefficient was proposed. The

equations that remained with a non-significant coefficient or presented more than one non-significant coefficient were disregarded for the rest of the study.

The accuracy of all models was evaluated by means of graphical analysis of the residues to identify the possible systematic errors, standard error of the estimate in percentage ($S_{yx}\%$) and adjusted determination coefficient ($R^2_{aj}$) [55].

Since the database contained repeated measurements of each tree—that is, hierarchical data—within the tree, the variance was heterogeneous and the correlation between the residues violated the basic assumptions [56]; therefore, we did not consider the regression conditions for the selection of equations.

### 2.6. Volume

The total volume of each tree was calculated by integrating the Clark III et al. model [30].

$$
v_j = 0.00007854 \left\{
\begin{array}{l}
I1 * d_j^2 * \left[ (1 - G * W) * (U1 - L1) + W * \dfrac{\left(\frac{1-L1}{h_j}\right)^{\beta_0} * (h_j - L1) - \left(1 - \frac{U1}{h_j}\right)^{\beta_0} * (h_j - U1)}{\beta_0 + 1} \right] + \\[4ex]
I2 * I3 * \left[ T * (U2 - L2) + Z * \dfrac{\left(\frac{1-L2}{h_j}\right)^{\beta_1} * (h_j - L2) - \left(1 - \frac{U2}{h_j}\right)^{\beta_1} * (h_j - U2)}{\beta_1 + 1} \right] + \\[4ex]
I4 * d_{5.3j}^2 * \left[
\begin{array}{l}
\beta_5 * (U3 - L3) - \beta_5 * \dfrac{(U3-5.3)^2 - (L3-5.3)^2}{h_j - 5.3} + \dfrac{\beta_5}{3} * \dfrac{(U3-5.3)^3 - (L3-5.3)^3}{(h_j-5.3)^2} + \\[3ex]
I5 * \dfrac{1}{3} * \dfrac{\frac{\beta_4 - 1}{\beta_4^2} * \left(\beta_4 * (h_j - 5.3) - (L3 - 5.3)\right)^3}{(h_j - 5.3)^2} - \\[3ex]
I6 * \dfrac{1}{3} * \dfrac{\frac{\beta_4 - 1}{\beta_4^2} * \left(\beta_4 * (h_j - 5.3) - (U3 - 5.3)\right)^3}{(h_j - 5.3)^2}
\end{array}
\right]
\end{array}
\right\}
$$

where $v_j$—total volume (m$^3$) of the tree $j$; $d_j$—diameter at breast height with bark (cm) measured for each tree $j$; $h_j$—height (m) of each tree $j$; $d_{5.3j}$—diameter with bark measured (cm) at a height of 5.3 m for each tree $j$; L—lower height of interest (m), L = 0; U—upper height of interest (m), U = $h_j$. G, W, X, Y, Z, T, L1, U1, L2, U2, L3 and U3—combined variables of the integral Clark III et al. model [30]: $G = (1 - 1.3/h_j)^{\beta_0}$, $W = \left(\beta_2 + \beta_3/d_j^3\right)/(1 - G)$, $X = (1 - 1.3/h_j)^{\beta_1}$, $Y = (1 - 5.3/h_j)^{\beta_1}$, $Z = \left(d_j^2 - d_{5.3j}^2\right)/(X - Y)$, $T = d_j^2 - Z * X$, L1—maximum of L and 0, U1—minimum of U and 1.3, L2—maximum of L and 1.3, U2—minimum of U and 5.3, L3—maximum of L and 5.3 and U3—minimum of U and $h_j$. $I1$, $I2$, $I3$, $I4$, $I5$ and $I6$—indicator variables of the integral Clark III et al. model [30]: $I1 = 1$ if $L < 1.3$, otherwise $I1 = 0$; $I2 = 1$ if $L < 5.3$, otherwise $I2 = 0$; $I3 = 1$ if $U > 1.3$, otherwise $I3 = 0$; $I4 = 1$ if $U > 5.3$, otherwise $I4 = 0$; $I5 = 1$ if $(L_3 - 5.3) < \beta_4(h_j - 5.3)$, otherwise $I5 = 0$; $I6 = 1$ if $(U_3 - 5.3) < \beta_4(h_j - 5.3)$, otherwise $I6 = 0$.

## 3. Results and Discussion

### 3.1. Descriptive Study of the Stem Forms

Preliminary to the descriptive study of the FCP in the test, the adjustment of the Schöepfer model [40] was performed to generate the derivative functions and to identify the FCPs. From the derivatives, trees with three inflection points, and thus, three FCPs were verified. These points could be three real inflection points, such as tree 1; two real inflection points and one imaginary (FCP$_3$), as presented with tree 6; and one real inflection point and two imaginary ones, which was not verified in the present study (Figure 1).

The stem of the trees can present a very variable form, which changes according to the species and even within the same species. It is known that the form of the tree stem is influenced by several factors, such as the site, spacing, thinning and sociological position [57,58]. In this sense, the identification of the change points has great theoretical value in the definition of mathematical expressions that describe the stem form.

It was verified in the study that the trees with the smallest DBH classes presented a greater distance between the FCP and the occurrence of the three FCPs (Table 5).

In trees of the medium stratum or dominated in dense stands, that is, trees that submitted to the competition, more cylindrical stems are expected with few FCPs [37]. However, *Tectona grandis* has vigorous branching, forming open globose crown and a superficial root system, allowing the stem to be endowed with tabular roots or not [59], and these characteristics may be intensified by the conditions of the settlement. The trees used in the study showed a wide variation of management regimes. Thus, under such characteristics, segmented models should result in better estimates of the stem dimensions for the species.

When analyzing the overall mean of the FCPs, it was verified that they were distributed proportionally along the stem. Up to 28% of the height, the lower portion of the stem could be described using a neiloid, which is a form that decreases strongly due to the effect of the tabular roots and is usually observed in the lower fifth of the trees [39]. The intermediate portion, from 28% to 57% of the height, could be described using a paraboloid, and from 57% to 73% of the height, still using a paraboloid or mild neiloid, and it was possible to characterize the ramification points of the open globose crown of the species. The apex, above 73% of the height, could be represented using a quadratic parable or a cone.

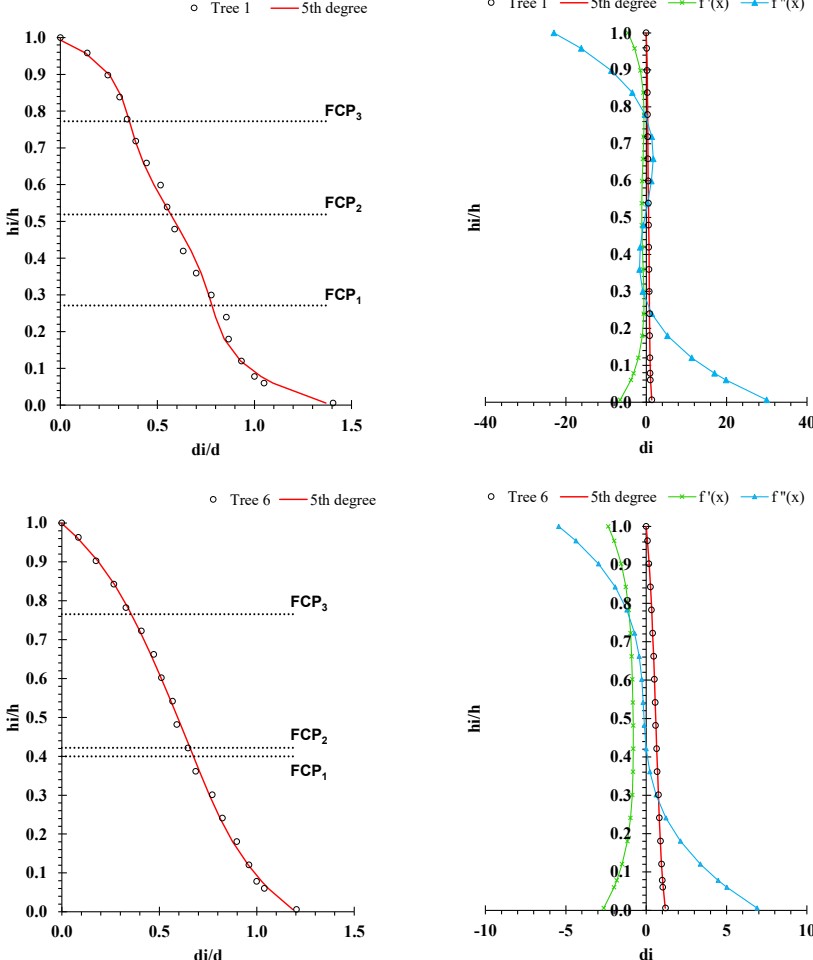

**Figure 1.** Medium profile derived from the 5th-degree function of Schöepfer [40] and form change points (FCPs) of the stems of *Tectona grandis* L.f., central-west region, Brazil. Legend: $h_i/h$−ratio between height at position i ($h_i$) and height (h); di−diameter (cm) with bark at position i obtained by the function x and its derivatives f′(x) and f″(x); di/d−ratio between diameter at position i ($d_i$) and diameter at breast height (d); f′(x)−first derivative of f(x); f″(x)−second derivative of f(x).

**Table 5.** Minimum, average and maximum DBH values and the total height of the form change points of the stem in each diameter class using the model of Schöepfer [40] for *Tectona grandis* L.f., central-west region, Brazil.

| Class DBH | Average Values in Each Diameter Class—Schöepfer Model (1966) [40] | | | | | | | | |
|---|---|---|---|---|---|---|---|---|---|
| | $FCP_1$ | $FCP_2$ | $FCP_3$ | $y1(d_i)$ | $x1(h_i)$ | $y2(d_i)$ | $x2(h_i)$ | $y3(d_i)$ | $x3(h_i)$ |
| 1 | 0.26 | 0.50 | 0.69 | 11.01 | 3.6 | 8.25 | 6.9 | 7.16 | 9.5 |
| 2 | 0.28 | 0.55 | 0.76 | 14.00 | 4.8 | 11.20 | 9.4 | 6.03 | 13.1 |
| 3 | 0.28 | 0.57 | 0.73 | 16.37 | 5.1 | 14.08 | 10.4 | 7.91 | 13.4 |
| 4 | 0.29 | 0.61 | 0.76 | 21.21 | 6.1 | 18.79 | 13.0 | 11.04 | 16.1 |
| 5 | 0.29 | 0.64 | 0.79 | 24.21 | 6.5 | 18.14 | 14.4 | 9.69 | 17.9 |
| 6 | 0.27 | 0.54 | 0.73 | 28.32 | 7.0 | 22.63 | 14.3 | 13.59 | 19.2 |
| 7 | 0.27 | 0.54 | 0.76 | 31.11 | 7.2 | 26.98 | 14.6 | 12.53 | 19.9 |
| 8 | 0.27 | 0.56 | 0.72 | 33.63 | 7.3 | 23.25 | 15.3 | 15.56 | 19.5 |
| 9 | 0.31 | 0.65 | 0.69 | 33.02 | 8.8 | 41.97 | 19.1 | 18.34 | 19.6 |
| 10 | 0.30 | 0.57 | 0.68 | 40.82 | 7.7 | 57.26 | 16.0 | 24.11 | 17.2 |
| Minimum | 0.26 | 0.50 | 0.68 | 11.01 | 3.6 | 8.25 | 6.9 | 6.03 | 9.5 |
| Maximum | 0.31 | 0.65 | 0.79 | 40.82 | 8.8 | 57.26 | 19.1 | 24.11 | 19.9 |
| Mean | 0.28 | 0.57 | 0.73 | 25.37 | 6.4 | 24.26 | 13.3 | 12.59 | 16.5 |

DBH—diameter at breast height; $FCP_n$—form change point *n*; y1—diameter corresponding to $FCP_1$, in cm; x1—height corresponding to the $FCP_1$, in m; y2—diameter corresponding to $FCP_2$, in cm; x2—height corresponding to the $FCP_2$, in m; y3—diameter corresponding to $FCP_3$, in cm; x3—height corresponding to the $FCP_3$, in m.

### 3.2. Adjustment of the Taper Models without Data Stratification

In general, single, segmented and variable-exponent equations presented adjustments to the species data (Table 6). The exceptions were the equations of Kozak et al. (b) (1969) [25], Kozak et al. (1969) (c) [25], Demaerschalk (d) [43], Demaerschalk (e) [43] and Parresol et al. [29], which did not converge, in addition to the equations of Demaerschalk (b) [43] and Cao et al. [45] (rewritten by Byrne; Reed [46]), which presented a singular Hessian matrix; these equations were discarded from the study.

**Table 6.** Parameters estimated in the adjustment of the taper models without stratification of *Tectona grandis* L. f. data, central-west region, Brazil.

| Model | Estimated Parameters | | | | | | | | | |
|---|---|---|---|---|---|---|---|---|---|---|
| | $\beta_0$ | $\beta_1$ | $\beta_2$ | $\beta_3$ | $\beta_4$ | $\beta_5$ | $\beta_6$ | $\beta_7$ | $\beta_8$ | $\beta_9$ |
| Single equation | | | | | | | | | | |
| Schöepfer [40] | 1.277 * | −5.658 * | 24.241 * | −50.413 * | 46.499 * | −15.931 * | | | | |
| Kozak et al. (a) (1969) [25] | | −2.219 * | 1.025 * | | | | | | | |
| Ormerod [44] | | 0.841 * | | | | | | | | |
| Demaerschalk [42] | 0.182 * | 0.913 * | 0.888 * | −0.895 * | | | | | | |
| Demaerschalk (a) [43] | 1.185 * | 1.778 * | | | | | | | | |
| Demaerschalk (c) [43] | 8686.5 * | 40.665 * | 1.089 * | 1.658 * | | | | | | |
| Segmented | | | | | | | | | | |
| Max and Burkhart [28] | 0.087 * | 0.713 * | −2.936 * | 1.411 * | 113.1 * | −1.203 * | | | | |
| Clark III et al. [30] | 40.875 * | 8.587 * | 0.898 * | 928.1 * | 0.655 * | 1.806 * | | | | |
| Variable-exponent | | | | | | | | | | |
| Kozak (1988) [36] | 1.091 * | 0.962 * | 0.999 * | 0.321 * | −0.177 * | −0.394 ns | 0.292 * | 0.055 * | | |
| Modified Kozak (1988) [36] | 1.086 * | 0.963 * | 0.999 * | 0.481 * | −0.215 * | | 0.083 * | 0.057 * | | |
| Kozak (a) (2004) [31] | 1.331 * | 0.981 * | 0.522 * | −0.048 ns | 0.024 * | −0.251 * | | | | |
| Modified Kozak (a) (2004) [31] | 1.345 * | 0.977 * | 0.511 * | | 0.025 * | −0.268 * | | | | |
| Kozak (b) (2004) [31] | 1.107 * | 0.953 * | 0.021 ns | 0.759 * | −1.287 * | −0.009 * | 1.034 * | 2.520 * | 0.145 * | −1.573 * |
| Modified Kozak (b) (2004) [31] | 1.140 * | 0.963 * | | 0.755 * | −1.322 * | −0.009 * | 1.010 * | 2.797 * | 0.149 * | −1.567 * |
| Bi [32] | 28.51 * | −13.49 * | −2.129 * | −15.034 * | 1.045 * | −14.99 * | 9.758 * | | | |
| Lee et al. [47] | 1.700 * | 0.907 * | 4.123 * | −5.715 * | 2.853 * | | | | | |

ns—not significant at the 5% error probability level; *—significant at the 5% error probability level.

The singular Hessian matrix occurs when the second partial derivatives with respect to the model parameters are negative and, therefore, make the matrix non-invertible; hence,

it is not possible to calculate the variance matrix. This "singular" state occurs for some combinations of datasets and models, and no computational process can reverse it, given the model and data chosen, since the desired inverse does not exist [60].

Only the variable-exponent equations, namely, Kozak (1988) [36], Kozak (a) (2004) [31] and Kozak (b) (2004) [31], presented coefficients that were not significant at the 5% probability of error and were not considered for the rest of the study. Modifications were proposed to the same equations, excluding the respective non-significant coefficients, which resulted in the significance of the remaining coefficients.

In general, the estimates were more accurate in the segmented and variable-exponent equations, followed by the single equations (Table 7). With this result, the hypothesis was confirmed since the segmented functions were more efficient at describing the taper of *Tectona grandis* stems.

**Table 7.** Adjustment and precision statistics for the taper models without stratification of the *Tectona grandis* L. f. data, central-west region, Brazil.

| Model | SQRes | $S_{yx}\%$ | $R^2_{aj.}$ |
|---|---|---|---|
| Single equation | | | |
| Schöepfer [40] | 11,378.8 | 9.63 | 0.97 |
| Kozak et al. (a) (1969) [25] | 22,940.1 | 13.67 | 0.95 |
| Ormerod [44] | 24,617.8 | 14.16 | 0.94 |
| Demaerschalk [42] | 22,492.0 | 13.54 | 0.95 |
| Demaerschalk (a) [43] | 23,764.0 | 13.92 | 0.94 |
| Demaerschalk (c) [43] | 18,077.2 | 12.14 | 0.96 |
| Segmented | | | |
| Max and Burkhart [28] | 10,658.6 | 9.33 | 0.97 |
| Clark III et al. [30] | 6447.5 | 7.25 | 0.98 |
| Variable-exponent | | | |
| Modified Kozak (1988) [36] | 8906.2 | 8.53 | 0.98 |
| Modified Kozak (a) (2004) [31] | 11,415.0 | 9.65 | 0.97 |
| Modified Kozak (b) (2004) [31] | 9054.2 | 8.60 | 0.98 |
| Bi [32] | 18,587.1 | 12.32 | 0.96 |
| Lee et al. [47] | 11,378.8 | 9.63 | 0.97 |

SQRes—sum of residual squares; $S_{yx}\%$—standard estimate error as a percentage; $R^2_{aj.}$—adjusted coefficient of determination.

The results of the $S_{yx}\%$ statistics, relative to the mean error of the estimates, were less than 15% for all the equations, with the most accurate mean values obtained in the segmented (7.25%) and variable-exponent (8.53%) models. Regarding the statistics, $R^2_{aj.}$ values greater than 94% were achieved for all the equations, which provided an indicator of appropriate estimates of the $d_i$ along the stem. For other species, Müller et al. [61] and Môra [62] using the Clark III et al. models [30], as well as Max and Burkhart [28], Demaerschalk [42], Kozak et al. (b) (1969) [25] and/or Schöepfer [40], also found better estimates in segmented equations relative to single equations.

The single equation models combine efficiency with the simplicity of application [63] but presented a significant bias in the estimate of diameters close to the soil, as well as in some other parts of the tree (Figures 2 and 3). The difficulties shared by many taper functions are the existence of a high degree of bias in predicting the diameter over some portions of the stem [32]. The greatest errors occurred at the ends of the stem, where the residuals of the single equations varied, on average, by −12 to 22 cm. The segmented functions presented more uniform distribution of the residues. However, all the equations showed some tendency along the estimation line, especially the tendency to underestimate the largest diameters.

More detailed graphical analysis of the distribution of residues for equations that presented better adjustment and precision statistics, namely, Clark III et al. [30], modified Kozak (1988) [36] and modified Kozak (b) (2004) [31], which presented $R^2_{aj.}$ values greater than or equal to 0.98 and $S_{yx}\%$ less than or equal to 9% with the use of variable-exponent

functions (modified Kozak (1988) [36] and modified Kozak (b) (2004) [31]), showed that it was not necessary to develop segmented taper functions for different parts of the stem in order to reduce the bias. This approach may allow for a lower degree of bias and greater accuracy in the taper predictions [36,53,64,65]. However, the segmented model proposed by Clark III et al. [30] is a combination of the best attributes of the Schlaegel and Max and Burkhart models, and it is capable of explaining the differences in taper by form classes, and as a consequence, it can present estimates of diameters more accurately than the others.

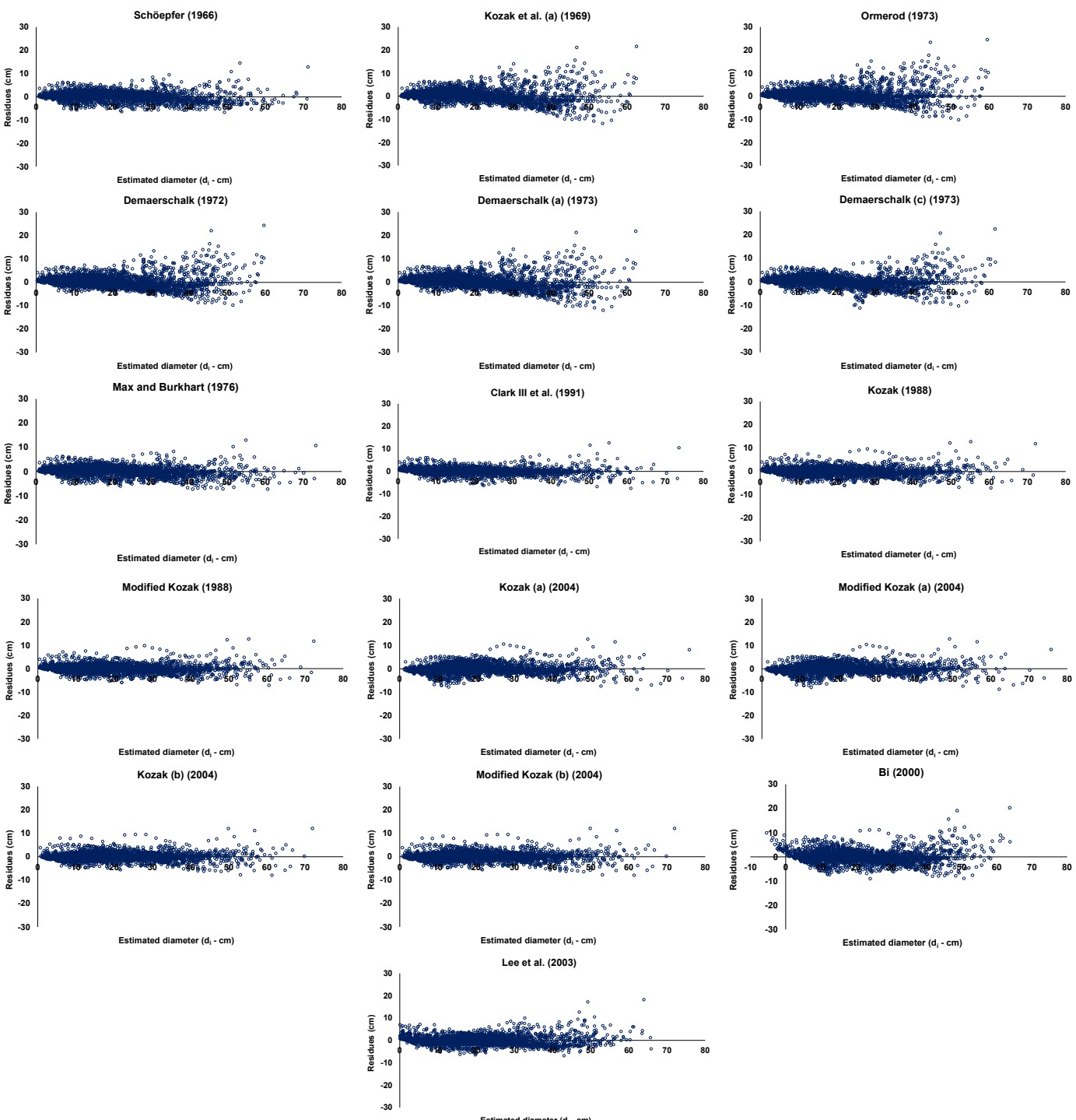

**Figure 2.** Residues from the estimates of diameters along the stem ($d_i$) using the taper equations without stratification of the *Tectona grandis* L.f. data., central-west region, Brazil [25,28,30–32,36,40,42–44,47].

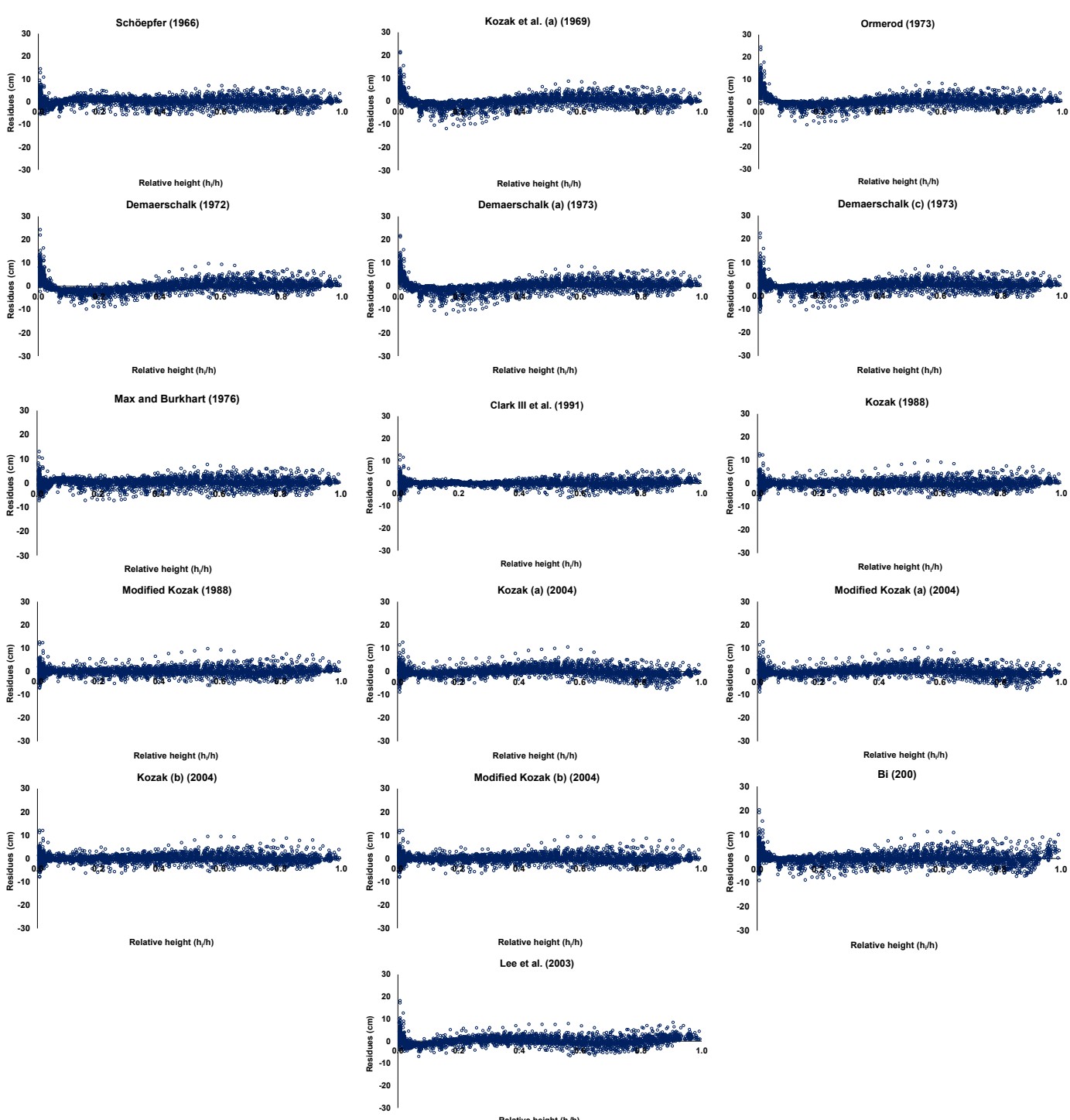

**Figure 3.** Residues from the estimates of diameters in the function of the relative height ($h_i/h$) when using the taper equations without stratification of *Tectona grandis* L.f. data., central−west region, Brazil [25,28,30−32,36,40,42−44,47].

The equation of Clark III et al. [30] presented errors with a variation between −7.55 to 12.66 cm, but with a more homogeneous distribution of residues in the mean portion of the stem when compared with those generated using the modified Kozak (1988) [36] and modified Kozak (b) (2004) [31] equations (Figure 3). The model of Clark III et al. [30] was the most accurate because it represented the differences in the form of a stem when the diameter at 5.3 m was measured and when compared with the other segmented models (Figure 3). In addition to properly estimating volumes at heights above 5.3 m, it offered

the best volume estimates at the base of the stem, which are quantities that had the largest dimensions and the highest added value [30].

When the diameter at 5.3 m is not available or measured, and it is not possible to estimate it using regression, the modified Kozak (1988) [36] equation can be used to estimate the diameters along the stem. However, the modified Kozak (1988) [36] equation has a higher degree of bias, especially in the middle part of the stem (Figures 2 and 3).

Unsatisfactory estimates at the upper extremes (85% to 95% of the height) and lower (1% of the height) were also reported by Favalessa et al. [66] in a study with *Tectona grandis* using segmented and single equation models. The authors observed similar errors at the extremities and, despite the tendency to underestimate the diameters at 95% of the height, recommended the Clark III et al. model [30] to accurately describe the diameters at the base of the stems where the greatest irregularities were found.

### 3.3. Stratification as a Variability Reducer in Models Adjustment

The generalization of the database increases the variability of estimates, which implies less accurate estimates when contrasted with the estimates of stratified adjustments. The recommendation of adjustment of diameter class taper functions is consistent with the results found by Fischer [67] and Assis et al. [68] for *Pinus taeda*; by Müller [69] for *Eucalyptus grandis* and Favalessa et al. [66] for *Tectona grandis*; using the form quotient class from Girard, Johnson and Hohenadl found by Souza et al. [50] for *Pinus taeda*; and by age class found by Friedl [70] and Figueiredo Filho et al. [71] for stands of *Araucaria angustifolia* and Kohler et al. [72] for *Pinus taeda*.

The equation of Clark III et al. [30] converged for all DBH classes; however, for the first two classes, they presented a unique Hessian matrix problem, and the others presented many non-significant coefficients (Table 8). In general, this stratification provided results that were similar to the adjustment without stratification. For some classes, gains in adjustment and accuracy were not verified, such as in class 10, which presented $R^2_{aj.}$ of 0.99 and $S_{yx}\%$ of 9.30%; these results did not exceed the adjustment without stratification results ($R^2_{aj.} = 0.98$; $S_{yx}\% = 7.25\%$).

A similar performance was also verified by Favalessa et al. [66] when studying *Tectona grandis* in the same region, both for segmented and single equation models, where for some DBH classes, the values of $S_{yx}\%$ were higher and those of $R^2_{aj.}$ were similar to those of an adjustment without stratification. Souza et al. [50] described those estimates at points far from the base of the stem, showing that the separation of data into DBH classes had no effect, being even lower than the non-stratified data.

In general, both stratification results by parameter class "$r$" and by $f_{1.3}$ were better than those obtained via stratification by DBH class, with the model converging in all classes, and all coefficients were significant at the 5% error probability level (Table 8). Similarly, as in the stratification by DBH class, gains in the adjustment and accuracy were also not verified, with a maximum error reduction of 1.4%. With the analysis of the distributions of the variables $f_{1.3}$ and parameter "$r$" as a function of the DBH, there was no clear trend in the distribution of the values as a function of DBH; furthermore, the values of $f_{1.3}$ presented strong variability between classes (Figure 4). The absence of expressive gains in accuracy with both stratifications could be attributed to the tendency of values being distributed parallel to the abscissa axis.

Among the proposed stratifications, the class of $f_{1.3}$ presented better results, which was explained by the decrease in variation within the classes of $f_{1.3}$ since, for a single DBH class, there may be different stem forms. The form of the tree can be affected by growth and the environment [73]; therefore, trees of the same DBH class do not necessarily have the same geometric form since these classes are produced arbitrarily or according to product demands [50]. In contrast, $f_{1.3}$ take into account the DBH and height variations, and therefore, they better represent the respective forms.

**Table 8.** Parameters estimated in the adjustments of Clark III et al. model without stratification and with stratification by diameter class at breast height, parameter "*r*" and artificial form factor for *Tectona grandis* L.f. data, central-west region, Brazil.

| Class of DBH | N | $L_i$ | $L_s$ | $\beta_0$ | $\beta_1$ | $\beta_2$ | $\beta_3$ | $\beta_4$ | $\beta_5$ | SQRes | $S_{yx}\%$ | $R^2_{aj.}$ |
|---|---|---|---|---|---|---|---|---|---|---|---|---|
| 1 | 20 | 10 | 15 | 25.531 * | 4.445 * | 0.899 * | 501.7 ns | 0.929 * | 4.384 ns | 78.1 | 6.82 | 0.99 |
| 2 | 20 | 15 | 20 | 37.490 * | 5.288 * | 0.215 ns | 4440.9 * | 0.935 * | 3.927 * | 113.5 | 4.79 | 0.98 |
| 3 | 20 | 20 | 25 | 39.231 * | 7.109 * | 1.047 * | −289.4 ns | 0.701 * | 1.797 * | 247.6 | 5.51 | 0.97 |
| 4 | 20 | 25 | 30 | 52.951 * | 8.742 * | 0.834 * | 6780.6 ns | 0.834 * | 3.509 * | 886.3 | 6.95 | 0.98 |
| 5 | 20 | 30 | 35 | 46.319 * | 8.370 * | 1.000 * | 22981.6 * | 0.600 * | 1.880 * | 745.8 | 6.29 | 0.98 |
| 6 | 16 | 35 | 40 | 50.722 * | 10.062 * | 0.893 * | 8047.0 ns | 0.721 * | 1.964 * | 731.3 | 5.87 | 0.98 |
| 7 | 18 | 40 | 45 | 37.376 * | 7.378 * | 1.097 * | −15525.2 ns | 0.560 * | 1.465 * | 1470.9 | 7.14 | 0.98 |
| 8 | 9 | 45 | 50 | 30.956 * | 14.406 * | −0.489 ns | 132825.0 * | 0.590 * | 1.540 * | 466.1 | 5.32 | 0.99 |
| 9 | 2 | 50 | 55 | −3.137 ns | 15.709 * | 0.567 ns | −11842.3 ns | 0.591 * | 1.758 * | 52.6 | 4.25 | 0.99 |
| 10 | 3 | 55 | 60 | 34.699 * | 12.687 * | 3.099 ns | −381868.0 ns | 0.647 * | 1.899 * | 424.6 | 9.30 | 0.99 |
| No Stratification | | | | 40.874 * | 8.585 * | 0.898 * | 929.1 * | 0.655 * | 1.806 * | 6449.1 | 7.25 | 0.98 |
| Class of *r* | N | Parameter "*r*" $L_i$ | $L_s$ | $\beta_0$ | $\beta_1$ | $\beta_2$ | $\beta_3$ | $\beta_4$ | $\beta_5$ | SQRes | $S_{yx}\%$ | $R^2_{aj.}$ |
| 1 | 85 | 0.3 | 0.8 | 42.646 * | 9.684 * | 0.924 * | 1745.9 * | 0.726 * | 2.028 * | 4379.6 | 7.27 | 0.98 |
| 2 | 45 | 0.8 | 1.3 | 42.861 * | 8.045 * | 0.877 * | 676.7 * | 0.551 * | 1.635 * | 1595.8 | 6.58 | 0.99 |
| 3 | 18 | 1.3 | 1.8 | 17.687 * | 5.595 * | 0.433 * | 1846.7 * | 0.589 * | 1.433 * | 177.4 | 5.82 | 0.99 |
| No Stratification | | | | 40.874 * | 8.585 * | 0.898 * | 929.1 * | 0.655 * | 1.806 * | 6449.1 | 7.25 | 0.98 |
| Class of $f_{1.3}$ | N | $f_{1.3}$ $L_i$ | $L_s$ | $\beta_0$ | $\beta_1$ | $\beta_2$ | $\beta_3$ | $\beta_4$ | $\beta_5$ | SQRes | $S_{yx}\%$ | $R^2_{aj.}$ |
| 1 | 32 | 0.30 | 0.38 | 47.673 * | 11.816 * | 0.841 * | 4840.6 * | 0.579 * | 1.468 * | 1600.2 | 5.95 | 0.99 |
| 2 | 100 | 0.38 | 0.46 | 39.092 * | 7.531 * | 0.925 * | 513.1 * | 0.697 * | 2.071 * | 3659.3 | 6.96 | 0.98 |
| 3 | 16 | 0.46 | 0.54 | 23.057 * | 4.215 * | 0.761 * | 1166.7 * | 0.957 * | 9.291 * | 415.7 | 7.57 | 0.98 |
| No Stratification | | | | 40.874 * | 8.585 * | 0.898 * | 929.1 * | 0.655 * | 1.806 * | 6449.1 | 7.25 | 0.98 |

ns—not significant at the level of 5% probability of error; *—significant at the level of 5% probability of error; N—absolute frequency observed in diameter class *i*; DBH—diameter at breast height, in cm; *r*—form parameter "*r*"; $f_{1.3}$—artificial form factor; $L_i$—class lower limit; $L_s$—class upper limit.

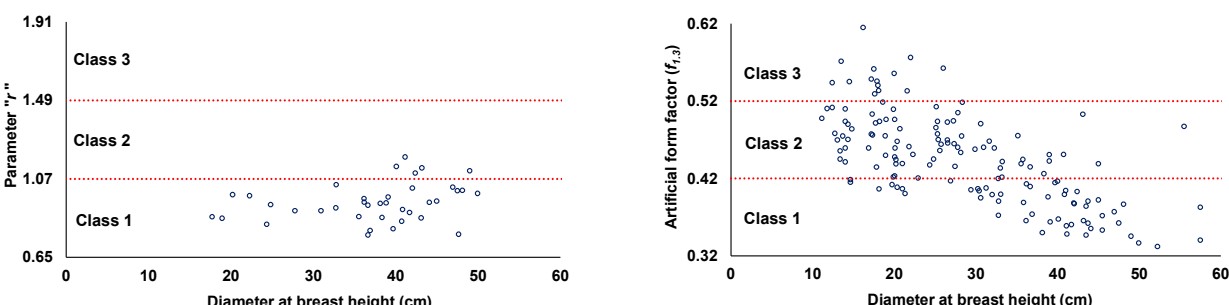

**Figure 4.** Stratification of the variables "*r*" and artificial form factor according to the diameter at breast height for *Tectona grandis* L.f., central-west region, Brazil.

Even though stratification in class $f_{1.3}$ presented gains in adjustment with the equation by Clark III et al. [30], this cannot be proposed as a more appropriate strategy. Based on the difficulty in classifying $f_{1.3}$ of the trees in the field, the $S_{yx}\%$ reductions did not justify the use of stratification.

### 3.4. Proposed Modifications to the Clark III et al. Model (1991)

Considering all the modifications implemented in the Clark III et al. equation [30], except for modification 3, which used the diameter at 6.4 m high ($d_{6.4j}$) instead of the

diameter at 5.3 m ($d_{5.3j}$) and the occurrence of a singular Hessian matrix, the decrease in $S_{yx}\%$ was lower than 0.2% (Table 9). The exclusion of components $\beta_3/d_j{}^3$ and $\beta_2 + \beta_3/d_j{}^3$ changed the description of the lower portion of the stem, reducing the equation flexibility. When comparing $S_{yx}\%$ and $R^2_{aj.}$ statistics with those calculated for the equation originally proposed by Clark III et al. [30], it was found that modifications 1 and 2 were not efficient at predicting the $d_{ij}$ along the stem for *Tectona grandis*.

**Table 9.** Estimated parameters and adjustment and precision statistics obtained in the adjustment of Clark III et al.'s model and after the proposed modifications of the same model for *Tectona grandis* L.f., central-west region, Brazil.

| Model | Estimated Parameters | | | | | | | SQRes | $S_{yx}\%$ | $R^2_{aj.}$ |
| | $\beta_0$ | $\beta_1$ | $\beta_2$ | $\beta_3$ | $\beta_4$ | $\beta_5$ | $\beta_6$ | | | |
|---|---|---|---|---|---|---|---|---|---|---|
| Clark III et al. [30] | 40.875 * | 8.587 * | 0.898 * | 928.2 * | 0.655 * | 1.806 * | | 6449.6 | 7.25 | 0.98 |
| Modification 1 | 41.837 * | 8.587 * | 0.943 * | | 0.655 * | 1.806 * | | 6500.3 | 7.28 | 0.98 |
| Modification 2 | 47.265 * | 8.587 * | | | 0.655 * | 1.806 * | | 6512.1 | 7.29 | 0.98 |
| Modification 3 | | | | Hessian matrix | | | | | | |
| Modification 4 | 40.875 * | 7.799 * | 0.898 * | 928.1 * | 0.603 * | 1.719 * | | 6378.8 | 7.21 | 0.98 |
| Modification 5 | 40.875 * | 8.587 * | 0.898 * | 928.1 * | 0.628 * | 1.707 * | 0.083 * | 6358.2 | 7.20 | 0.98 |
| Modification 6 | 40.875 * | 7.799 * | 0.898 * | 928.1 * | 0.548 * | 1.565 * | 0.348 * | 6118.4 | 7.07 | 0.99 |
| Modification 7 | 40.875 * | 8.587 * | 0.898 * | 928.1 * | 0.610 * | 1.646 * | 0.017 * | 6252.9 | 7.14 | 0.99 |
| Modification 8 | 40.875 * | 7.799 * | 0.898 * | 928.1 * | 0.548 * | 1.564 * | 0.020 * | 6139.3 | 7.08 | 0.99 |

\*—significant at the 5% error probability level; SQRes—sum of residue squares; $S_{yx}\%$—standard estimate error in percentage; $R^2_{aj.}$—adjusted coefficient of determination.

The modifications made based on the study of the FCPs, namely, modification 4, which used $d_{0.28h}$ instead of $d_{5.3j}$, and modification 5, which included the coefficient $\beta_6$ with the $d_{0.73h\%}$ in the Clark III et al. model [30], provided a reduction in the $S_{yx}\%$ by only 0.05%. With the graphical analysis of the residue dispersion with respect to the estimated $d_i$ (Figure 5), no significant gain was found to justify these changes in the measuring positions along the tree stems.

Modification 6, which replaces $d_{5.3j}$ with $d_{0.8h}$ and added $\beta_6 \times d_{0.73h}$, achieved a slight improvement in the adjustment statistics. Upon segmenting the Clark III et al. model [30] in the actual positions of form change of the species, the equation described the stem more appropriately, although with a slight improvement. However, when the residue dispersion charts were analyzed, no difference was found between the original Clark III et al. model [30] and the models with modifications 2 to 6. When using the inverse of the $d_i/h$ ratio as the stem descriptor, i.e., modifications 7 and 8, there was a slight improvement in the precision. Modification 8, namely, the substitution of $d_{5.3j}$ with $d_{0.28h} + \beta_6 \times 1/hd_j$, brought a 0.17% reduction in the calculated result. However, reductions in the error of each magnitude did not justify the implementation of any of the changes.

Standing tree scaling is an activity that demands time and, consequently, it is a burden to update the forest inventory. Therefore, keeping only the fixed 5.3 m measurement is more feasible than introducing the need for the measurement of new dimensions in the field. However, if the object tree has a total height lower than 7.5 m ($h_{0.28h} \leq 2.1$ m), it is possible to obtain $d_{0.28h}$ without using sophisticated equipment such as electronic dendrometers and/or without climbing the tree, making it feasible to use modification 8 to estimate diameters along the stem.

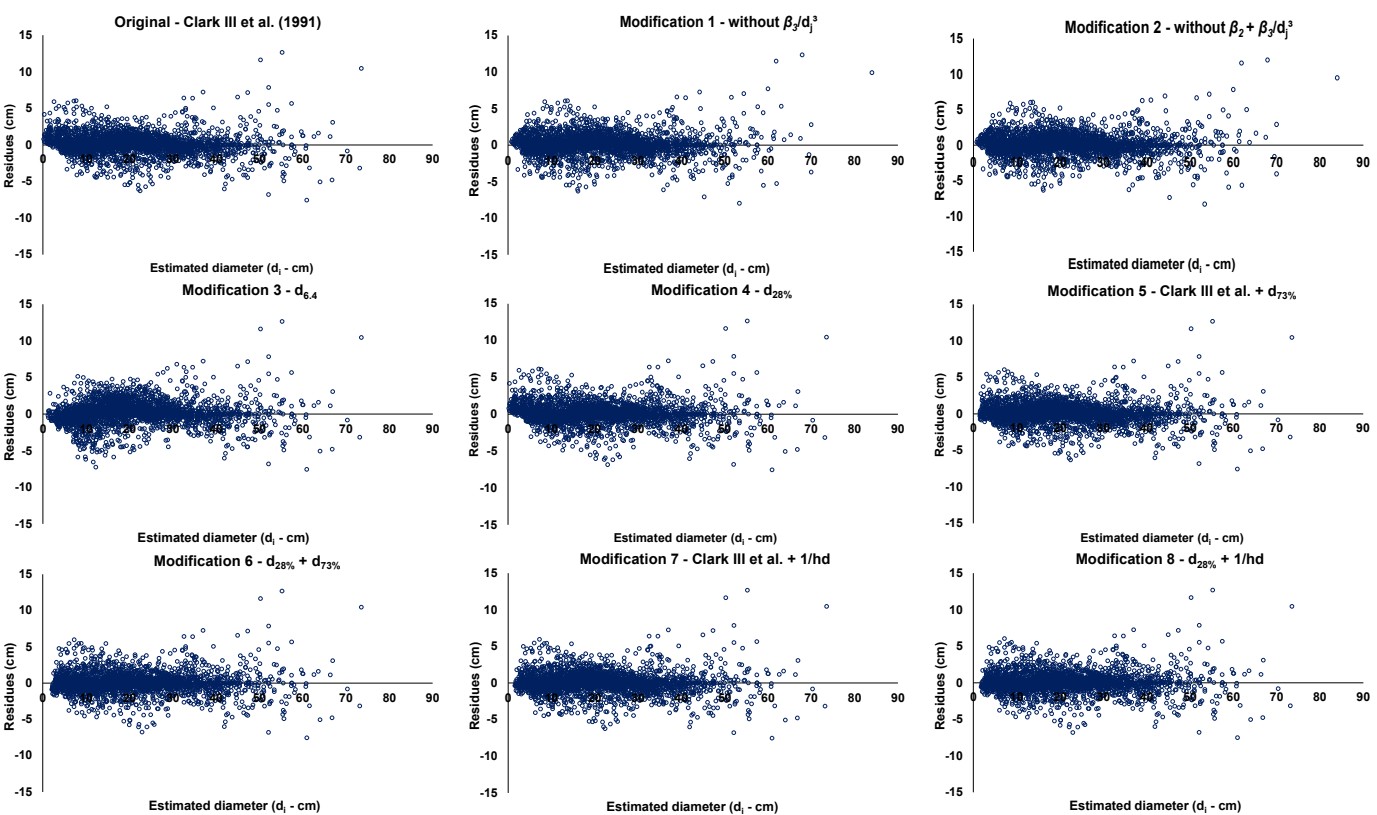

**Figure 5.** Residues of the estimates of relative diameters using the equations of Clark III et al. and the proposed modifications to the *Tectona grandis* L.f. data, central−west region, Brazil [30].

*3.5. Volume*

The volume calculated using the integration of the Clark III et al. model [30] showed an accuracy of 8.2%, but with a tendency to underestimate the lower volumes and overestimate the higher volumes compared with the actual volume (Figure 6).

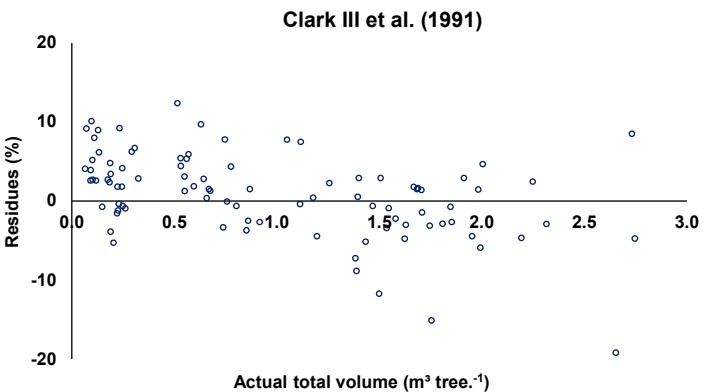

**Figure 6.** Residue of the total volume estimate using the Clark III et al. integral model as a function of the actual total volume for *Tectona grandis* L.f. data without stratification, central−western region, Brazil [30].

These results were less accurate than the models evaluated by Favalessa et al. [66] for the same species, which also showed trends in underestimating the lowest volumes and overestimating the highest volumes along the midline, with an accuracy of 7.15% for the integral of the Clark III et al. equation [30]. However, the authors used data with a smaller variation range (22.9 to 44 cm) from only one site and with the same management regime.

Lanssanova et al. [21], by adapting the nonlinear mixed-effect tapered stem model for *Tectona grandis* in the midwestern region of Brazil and using different heights as calibration, obtained root mean square error (RMSE%) values of volume estimates ranging from 6.61% to 20.82%. Thus, in addition to estimating the diameters with good accuracy, the integral of Clark III et al. [30] also provided adequate volume estimates.

## 4. Conclusions

Based on the sample data and the results of the Schöepfer equation [40], the form change points of the *Tectona grandis* stem occurred at 28%, 57% and 73% of the height of the stem, corresponding, on average, to the absolute positions of 6.4 m, 13.3 m and 16.5 m.

While testing the use of single, segmented and variable-exponent taper equations in the description of the stem form, the segmented equation by Clark III et al. [30], with the use of data without stratification in classes, was found to be the most appropriate to estimate the diameters along the stem and the volume of *Tectona grandis* trees in Mato Grosso.

When the diameter at 5.3 m was not available or measured, and it was not possible to estimate it, the modified Kozak equation (1988) [36] could be used to estimate the diameters along the stem, even with greater bias, especially in the middle part of the stem. Moreover, if the object tree had a total height of lower than 7.5 m ($h_{0.28h} \leq 2.1$ m), it was possible to use modification 8 of the model by Clark III et al. [30] without the aid of sophisticated equipment and/or without climbing the tree to estimate the diameters along the stem. Care must be taken when using these models on other sites and predicting beyond the observed range of the tree size.

**Author Contributions:** Conceptualization, K.J.d.R. and C.M.C.F.; methodology, K.J.d.R. and C.A.G.F.; software, K.J.d.R.; validation, K.J.d.R., C.A.G.F. and F.D.F.; formal analysis, K.J.d.R. and C.M.C.F.; investigation, K.J.d.R.; resources, K.J.d.R. and S.F.C.; data curation, K.J.d.R. and S.F.C.; writing—original draft preparation, K.J.d.R.; writing—review and editing, K.J.d.R., C.A.G.F., C.M.C.F., S.F.C. and F.D.F.; visualization, K.J.d.R.; supervision, K.J.d.R. and C.A.G.F.; project administration, K.J.d.R., C.A.G.F. and C.M.C.F.; funding acquisition, S.F.C. All authors have read and agreed to the published version of the manuscript.

**Funding:** This research received no external funding.

**Institutional Review Board Statement:** Not applicable.

**Informed Consent Statement:** Not applicable.

**Data Availability Statement:** Not applicable.

**Conflicts of Interest:** The authors declare no conflict of interest.

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
