# Peer review of "Form and Volume of the Stem of Tectona grandis L.f. in the Central-WESTERN Region of Brazil"

_forests, doi:10.3390/f13111818_

Round 1

Reviewer 1 Report

Forests, manuscript forests-1928455-peer-review-v1 “Form and volume of the stem of Tectona grandis under different management regimes”

Taper models have become a standard way for volume estimation. As they can not only estimate total volume but partial volume. The authors present a study that focuses on form and volume of the stem under different management regimes. But in the context, there are not details about the management regimes. Readers may be more interested in what will happen on tree stem under different management regimes. Furthermore, the manuscript requires significant editing. It is a good idea to write short clear sentences. In many sentences I could not understand what the authors were trying to communicate. In addition, the presentation of the results and some parts of the Discussion section must be largely improved.

Major concerns:

1.      Lines 1-2: In the title, you mentioned the different management regimes. But in the context, there are not details about the management regimes.

2.      Lines 69-70: Please give the maximum and minimum values of diameter at breast height and total tree height.

3.      Lines 104-105: Why did you select the Schöepfer model? Please explain.

4.      Line 119: Kozak(1988) (Kozak [6] ) is a variable-exponent model, but it is classified as a segmented model

5.      Line 138: In the function gw=p*wr, where gw is the basal area of the stem at a distance w from the crown edge and p is the parameter. Is there something wrong with this function? I think the correct function should be gw=π*p*wr.

6.      Line 151: There are two errors in Clark III et al. [20] formula in Table 4 (on page 4). One error is that 0.5 is missing from the outermost curly bracket (0.5 marked in red in the screenshot below). Another mistake is that the square bracket is in the wrong place, which I've indicated with red arrows. Similarly, Modifications 1-8 in Table 4 have the same error. Errors in these formulas mean that the results after Section 3.4 of the paper may be incorrect.

7.      Lines 186-187: How did you define partial volume? Please give more details about the definition of partial volume. You should also give details on how to calculate total and partial volume by integration, and an accurate reference should be given.

8.      Lines 295-297: The Clark III et al. model showed more accuracy since it used diameter at 5.3 m to calibrate the model. When the diameter at 5.3 m was not available, which model should be used? Please also give the recommendation.

9.      Lines 394-408: The partial volume was not calculated as described in Lines 186-187.

10.   Lines 402-403: Models should be compared by estimating total and partial volume.

11.   Lines 410-412: The Clark model should be used for trees with total tree height over 5.3 m. Are all trees have total tree height over 5.3 m? Please give some details about the minimum values of total tree height on data description part.

12.   Please try to provide references in English rather than heavily citing references from other languages.

Author Response

First of all, the authors would like to thank you for your contribution. It was significant for the improvement of the manuscript.

1.1 Lines 1-2: In the title, you mentioned the different management regimes. But in the context, there are not details about the management regimes.

Attended, lines 1 and 2.

“Form and volume of the stem of Tectona grandis in the central-western region of Brazil”

1.2 Lines 69-70: Please give the maximum and minimum values of diameter at breast height and total tree height.

Attended, table 1.

The authors inserted the standard deviation.

1.3 Lines 104-105: Why did you select the Schöepfer model? Please explain.

Attended, lines 144-146.

The text was rewritten.

1.4 Line 119: Kozak (1988) (Kozak [6]) is a variable-exponent model, but it is classified as a segmented model

Attended, lines 158.

The text was rewritten.

“... – and segmented variable-exponent – Bi [32], Lee et al. [47] and Kozak (2004) [31] ...”

1.5 Line 138: In the function gw=p*wr, where gw is the basal area of the stem at a distance w from the crown edge and p is the parameter. Is there something wrong with this function? I think the correct function should be gw=π*p*wr.

Attended, line 176-178.

“The form parameter “r” was expressed by the function gw=p*wr, where gw is the basal area of the stem at a distance w from the crown edge and p is the parameter that describes the size of the rotation body, where: p=a*π, a is the curve inclination.”

1.6 Line 151: There are two errors in Clark III et al. [20] formula in Table 4 (on page 4). One error is that 0.5 is missing from the outermost curly bracket (0.5 marked in red in the screenshot below). Another mistake is that the square bracket is in the wrong place, which I've indicated with red arrows. Similarly, Modifications 1-8 in Table 4 have the same error. Errors in these formulas mean that the results after Section 3.4 of the paper may be incorrect.

Attended, table 4.

Typing error. The scripts are correct.

1.7 Lines 186-187: How did you define partial volume? Please give more details about the definition of partial volume. You should also give details on how to calculate total and partial volume by integration, and an accurate reference should be given.

Attended, lines 226-239.

The text was rewritten.

1.8 Lines 295-297: The Clark III et al. model showed more accuracy since it used diameter at 5.3 m to calibrate the model. When the diameter at 5.3 m was not available, which model should be used? Please also give the recommendation.

Attended, lines 353-356.

“When the diameter at 5.3 m is not available or measured, and it is not possible to estimate it by regression, the modified Kozak (1988) [36] equation can be used to estimate the diameters along the stem. However, the modified Kozak (1988) [36] equation has a higher degree of bias, especially in the middle part of the stem (Figures 2 and 3).”

1.9 Lines 394-408: The partial volume was not calculated as described in Lines 186-187.

Attended, lines 226-239 and 456-458.

The text was rewritten.

1.10 Lines 402-403: Models should be compared by estimating total and partial volume.

In this study the authors only calculated the total volume.

1.11 Lines 410-412: The Clark model should be used for trees with total tree height over 5.3 m. Are all trees have total tree height over 5.3 m? Please give some details about the minimum values of total tree height on data description part.

Attended, table 1.

The authors inserted the standard deviation.

1.12 Please try to provide references in English rather than heavily citing references from other languages.

Attended.

Reviewer 2 Report

Dear authors,

the manuscript could be improved with respect to the methodical description. Also, I think, the novelty and impact of the study could be strengthed.

Please refer to PDF for specific comments.

Kind regards.

Author Response

First of all, the authors would like to thank you for your contribution. It was significant for the improvement of the manuscript.

2.1 Line 16: non-segmented is simple taper functions?

Attended, line 18.

The text was rewritten.

“…, single equation, segmented and variable-exponent taper functions to describe the stem profile.”

2.2 Line 16: variable-form taper functions are commonly referred to as variable-exponent taper functions.

Attended, line 18.

The text was rewritten.

“… and variable-exponent taper functions to describe the stem profile.”

2.3 Line 52-53: Please quote the corresponding references here.

Attended. The text was rewritten.

2.4 Line 58-62: his paragraph completely repeats the first sentence of the abstract.

Attended. The text was rewritten.

2.5 Line 82: from the soil??

Attended, line 114.

“… diameter at 1.30 m at breast height (dj).”

2.6 Line 87-88: what is adjustment database?

Attended, line 120.

The text was rewritten.

“The scaled trees were distributed in ten diameter classes at breast height (DBH) and were randomly selected to compose the adjustment database for taper models (Table 2).”

2.7 Line 105-108: I do not understand what you mean here.

Attended, lines 137-140.

The text was rewritten.

“The point where the change occurs in the form of this geometric figure is where the curve describing the profile of the stem changes of concavity, that is, the inflection point of the curve, mathematically obtained from the second derivative of the zero-equal function.”

2.8 Line 117, 119 and 120: To identify different Kozak functions, you should add the age.

Attended.

2.9 Line 136-137: Table 3 - Where did the data come from?

The data comes from the sample trees, as mentioned in lines 163-168.

The methodology for calculating the form factor was presented in lines 171-173; and for calculating r, lines 176-178.

Would the reviewer like the authors to enter values from the sample trees or mention Table 2?

We would just like to know how best to meet the reviewer's requirements.

2.10 Line 153-158: Parameter symbols are too vague. dj, dij, d5.3j, d0.28h, hj, hij, h0.73h

The authors have tried to follow the symbols proposed in reference articles such as Kozak (2004). For example, in Kozak (2004), page 508: hi = height from ground (m); di = inside bark diameter at hi height from ground (cm).

However, if the reviewer can propose a clearer way, it will be a pleasure to attend to it.

Kozak, A. My last words on taper equations. Forestry Chronicle 2004, 80, 507 – 515.  

2.11 Line 158: What are the values of IS, IB, IT and IM? When is 1 or 0?

Attended, lines 196-199.

“…IS, IB, IT and IM - connection points of the Clark III et al. model [30]: IS = 1 if dij2dj2, otherwise IS = 0; IB = 1 if dj2 > dij2F2, otherwise IB = 0; IT = 1 if F2 > dj2, otherwise IT = 0; IM = 1 if dj2 > ( -1)2F2, otherwise IM = 0; F = d5.3j2. For modifications 1, 2, 5 and 7: F = d5.3j2; for modification 3: F = d6.4j2; for modifications 4, 6 and 8: F = d0.28hj2.”

2.12 Line 192: What is FMP?

Attended, line 224: “FCP”.

Typing error.

2.13 Line 198-199: what is on the right? You should explain what negative numbers stands for and why the two trees are so different?

What is on the right are the values for the derivatives. The derivative of a function y = f (x) is the ratio of the infinitesimal increments of the function y and the variable x. The first derivative tells you about the slope of the function graph: f'(x) < 0 - decreasing; f'(x) > 0 - increasing.

The second derivative [f”(x)] is the derivative of the derivative of the function y=f(x) at a generic point of abscissa x, instantaneous rate of change. The sign of the second derivative of a function indicates the orientation of the concavity of its graph.

The inflection point is a point where the direction of the concavity changes, i.e., the point where the shape of the function changes (Formal Change Point – FCP). Consequently, it is a point where the second derivative changes sign, i.e., it is a point that corresponds to a zero second derivative.

The difference between the two trees is the number of shape change points, inflection points of the function. Tree 1 has 3 FCP points and tree 6 has only 2 FCP points. Tree 6 has two real FCP's and one imaginary FCP.

In this sense, the identification of the change points plays a great theoretical value in the definition of mathematical expressions that describe the stem form.

In lines 245-247, the authors present the differences between trees 1 and 6, figure 1.

Would the reviewer like the authors to present the differences between the trees and explain why the negative values in the figure title?

Figure 1 - The title of the abscissa x has been corrected.

2.14 Line 225: ????

By sectioning the stem of the tree into 5 parts, the lower fifth would be from the ground up to 20% of the stem. The term "lower fifth" was used by Prodan et al. (1997).

Would the reviewer like the authors to use another way to refer to this portion of the stem?

Prodan, M.; Peters, R.; Cox, F. Mensura florestal. IICA: San José, Costa Rica. 1997. 586 p.

2.15 Line 273: This residues are too big.

Lines 324-325.

Yes, the authors are reinforcing one of the difficulties that many taper functions have, especially the non-segmented ones, in estimating stem ends.

The text was rewritten.

2.16 Line 292-293: This error is too big.

The largest errors in the Clark III et al. (1991) equation were in estimating diameters from the ground to 10% of the stem height, -7.55 cm to 12.66 cm. Some trees that correspond to the study have DBH = ~60 cm, and for these, diameters of the first 10% of the tree's stem range from 84 cm to 69 cm. When converting into percentages, the residual of -7.55 cm corresponds to -14%; and the residual of 12.66 cm corresponds to 18%.

The authors understand the reviewer's concern regarding the accuracy of the equations, however, the authors bring some papers to clarify that the values obtained are good when compared to what is available in the literature for the species. Reinforcing that the results obtained by the authors were without stratification and without mixed effects calibration.

  1. Choochuen et al. (2021), evaluated calibration of a mixed-effect tapered stem model for teak cultivation in western Thailand and compared tree shape in the western and northern regions of the country, which included 8 age classes, and stratified the results into relative height and total height classes. The relative prediction errors of the estimate in percent (Syx%) were:

diâmetro - from 3.81% to 9.82%.

  1. Lanssanova et al. (2020), with calibration of a mixed-effect tapered stem model for tectona grandis in the central-western region of Brazil, obtained the following variations of root mean square errors (RMSE%) of diameter and volume estimates along the stem at different heights tested for mixed-effect model calibration:

diameter - 6.15% to 15.74%

  1. Seppänen and Mäkinen (2020), studying teak stands in Panama, obtained 9.9% Root Mean Square Error with an eighth degree polynomial to estimate diameter.

  1. Lanssanova et al. (2018), with non-linear equation fits in their basic formulations and with mixed effect in different scenarios, and for stratification in three portions of the total height, for Tectona grandis in the central-western region of Brazil, obtained standard errors of the estimate in percent (Syx%) for diameter estimates

No stratification - fixed effect: from 8.71% to 12.52%; mixed effect: from 7.68% to 8.19%.

With stratification - fixed effect: from 5.83% to 30.92%; mixed effect: from 5.53% to 25.76%.

  1. Pelisari et al. (2017), estimating the volume spatial variability for timber assortments and identify their spatial patterns on Tectona grandis stands in the central-western region of Brazil, obtained the following variations of root mean square errors (RMSE%) of diameter and volume estimates along the stem:

diameter - 7.7% to 8.3%

Choochuen, T.; Suksavate, W.; Meunpong, P. Development of a taper equation for teak (Tectona grandis L.f.) growing in Western Thailand. Environ. Nat. Resour. J. 2021, 19, 176-185. doi: 10.32526/ennrj/19/2020183.

Lanssanova, R.L.; Machado, S.A.; Orso, G.A.; Pelissari, A.L.; A Figueiredo Filho, A.; Silva, F.A. Calibration of a mixed-effect stem taper model for Tectona grandis. J.  Trop. For. Sci. 2020, 32, 341–348. doi: 10.26525/jtfs2020.32.4.341.

Seppänen P., Mäkinen A. Comprehensive yield model for plantation teak in Panama. Silva Fennica 2020, 54, 1-25. doi: 10.14214/sf.10309.

Lanssanova, L.R.; Machado, S.A.; Garrett, A.T.A.; Bonete, I.P.; Pelissari, A.L.; Figueiredo Filho, A.; Silva, F.A.; Ciarnoschi, L.D. Mixed-effect non-linear modelling for diameter estimation along the stem of Tectona grandis in mid-western Brazil. South. For. 2018, 1-7. doi: 10.2989/20702620.2018.1531279.

Pelissari, A.L.; Roveda, M.; Caldeira, S.F.; Sanquetta, C.R.; Dalla Corte, A.P.; Rodrigues, C.K. Geostatistical modeling of timber volume spatial variability for Tectona grandis L.F. precision forestry. Cerne 2017, 23, 115-122. doi: 10.1590/01047760201723012291.

2.17 Line 312: What's the meaning?

Attended, lines 368-369.

“… by form quotient class from Girard, Johnson and Hohenadl found by Souza et al. [50] for Pinus taeda; …”

2.18 Line 359: I don't think references should be added here.

Attended.

2.19 Line 395-396: The accuracy is only 8.2%?

The authors understand the reviewer's concern regarding the accuracy of the equations, however, the authors bring some papers to clarify that the values obtained are good when compared to what is available in the literature for the species. Reinforcing that the results obtained by the authors were without stratification and without mixed effects calibration.

  1. Choochuen et al. (2021), evaluated calibration of a mixed-effect tapered stem model for teak cultivation in western Thailand and compared tree shape in the western and northern regions of the country, which included 8 age classes, and stratified the results into relative height and total height classes. The relative prediction errors of the estimate in percent (Syx%) were:

volume - from 4.12% to 9.65%.

  1. Lanssanova et al. (2020), with calibration of a mixed-effect tapered stem model for tectona grandis in the central-western region of Brazil, obtained the following variations of root mean square errors (RMSE%) of diameter and volume estimates along the stem at different heights tested for mixed-effect model calibration:

volume - 6.61% to 20.82%

  1. Pelisari et al. (2017), estimating the volume spatial variability for timber assortments and identify their spatial patterns on Tectona grandis stands in the central-western region of Brazil, obtained the following variations of root mean square errors (RMSE%) of diameter and volume estimates along the stem:

volume - 12.8% to 26.8%

Choochuen, T.; Suksavate, W.; Meunpong, P. Development of a taper equation for teak (Tectona grandis L.f.) growing in Western Thailand. Environ. Nat. Resour. J. 2021, 19, 176-185. doi: 10.32526/ennrj/19/2020183.

Lanssanova, R.L.; Machado, S.A.; Orso, G.A.; Pelissari, A.L.; A Figueiredo Filho, A.; Silva, F.A. Calibration of a mixed-effect stem taper model for Tectona grandis. J.  Trop. For. Sci. 2020, 32, 341–348. doi: 10.26525/jtfs2020.32.4.341.

Pelissari, A.L.; Roveda, M.; Caldeira, S.F.; Sanquetta, C.R.; Dalla Corte, A.P.; Rodrigues, C.K. Geostatistical modeling of timber volume spatial variability for Tectona grandis L.F. precision forestry. Cerne 2017, 23, 115-122. doi: 10.1590/01047760201723012291.

2.20 Line 402: not 8.2%?

Attended, lines 463-467.

The text was rewritten.

2.21 Line 407-408: The accuracy is only 8.2% or 7.15%, it is not good.

Attended, lines 463-472.

The text was rewritten.

2.22 Line 412: The DBH and height of each tree are different, so I don't think can determine the absolute positions of the form changes points.

The authors understand and agree that each tree has a different DBH and height, but the absolute values ​​presented correspond to the average of 148 trees used in the present study. The authors did not generalize the FCP or the absolute values ​​for all trees of Tectona grandis.

Line 36: “Based on the sample data and the results of the Schöepfer equation [10], …”.

Reviewer 3 Report

The tree stem form and its modeling approach is a very important scientific issue in forestry research. To my opinion, the authors sufficiently tried to describe the stem form of trees of Tectona grandis L.f. However, there are some points that the authors must re-visit and issues that have to be addressed. Below I am giving my comments and suggestions.

Abstract:

  1. Lines 20 – 22: “FCP were determined……of the species.” Something is missing. Please re-write.

Introduction

Τhe introduction section needs to further be enriched by relative literature. There are no references to other related works, to form and volume generally, and specifically for the stem of Tectona grandis. Furthermore, there are no references to the methods that the authors are going to use. Four references are not enough evidence to support the Introduction section. Conclusively, this section should be substantially improved.

  1. Line 44. Delete the word “wood”
  1. Lines 48: Delete the word “and”

Materials and Methods

 Lines 65 – 66: Be clear about your sampling scheme. In order to support your sampling scheme, further details are needed. For example, which sampling design was used? Which was the size of your sample plots? Do you evaluate (under some statistical criteria) your choice of the plots’ size?

  1. Lines 75 – 80: “The scaling was performed…… d by the models.” Something is missing. It is difficult to understand what you have do
  2. Line 82: Which height? Do you mean the total height (ground to top of the tree?) or something else?
  3. Lines 84 – 85: Insert some information for the concept.
  4. Line 87: Why 25 trees were selected? This statement needs to be supported or by references or by some kind of statistical inverstigation
  5. Line 82: Beyond the reference, it would be informative, if you gave the equation of the Schöepfer function
  6. Line 121: What do you mean by “diameter (dij) variable was isolated” ? Please, explain.
  7. Lines 128 – 130: You have to support your statement by references.
  8. Statistical evaluation of the models: You have used non-linear regression analysis, in order to estimate the coefficient values of the models. In the preprocessing phase, do you test any of the prerequisites of the regression analysis? If you have made the testing, you have to describe your results.    

Results and discussion

  1. Lines 190 – 193: The statistical evaluation criteria of the fitted models, have to presented in the Materials and Methods section.  

Conclusions

According to the length of the paper, this section is too small. I am sure that more conclusions derived from this study.

Author Response

First of all, the authors would like to thank you for your contribution. It was significant for the improvement of the manuscript.

3.1 Lines 20 – 22: “FCP were determined……of the species.” Something is missing. Please re-write.

Attended.

The abstract was rewritten.

Introduction: Τhe introduction section needs to further be enriched by relative literature. There are no references to other related works, to form and volume generally, and specifically for the stem of Tectona grandis. Furthermore, there are no references to the methods that the authors are going to use. Four references are not enough evidence to support the Introduction section. Conclusively, this section should be substantially improved.

3.2 Line 44. Delete the word “wood”.

Attended.

3.3 Lines 48: Delete the word “and”.

Attended.

Materials and Methods

3.4 Lines 65 – 66: Be clear about your sampling scheme. In order to support your sampling scheme, further details are needed. For example, which sampling design was used? Which was the size of your sample plots? Do you evaluate (under some statistical criteria) your choice of the plots’ size?

Attended, lines 97-98.

“… Brazil (Table 1). The trees were randomly obtained in the final harvest of the stands. The climate of…”.

The manuscript is part of the experiments conducted by the first author during the course of graduate school to obtain the title of Doctor of Forestry Engineering. This first part of the study used trees obtained from final harvest, therefore, it is not relevant to present the sampling scheme adopted for monitoring the stand.

The ideal sampling intensity of the trees, per diameter class, was verified, considering the population as infinite and the maximum allowable error limit of 10% on average at 95% confidence probability, as recommended by Péllico Netto and Brena [8] (Table 2).

In any case, to clarify the reviewer:

  • Aguaçu and Soroteca farms adopted rectangular plots with fixed area of 600 m² and 900 m², respectively; and simple random sampling process with 1 sampling unit for 8.5 ha, on average.
  • Teca do Brasil and Teca do Jauru farms adopted rectangular plots with fixed area of 450 m² and 1000 m², respectively; and systematic sampling process with 1 sampling unit for 4.5 ha, on average.

3.5 Lines 75 – 80: “The scaling was performed…… d by the models.” Something is missing. It is difficult to understand what you have do

Attended, lines 106-112.

The text was rewritten.

3.6 Line 82: Which height? Do you mean the total height (ground to top of the tree?) or something else?

Attended, line 113-114.

“… at various heights of ground to top of the tree…”

3.7 Lines 84 – 85: Insert some information for the concept.

Attended, line 117-118.

“The dominant height (h100) was calculated by considering the concept of Assmann [37], average height of the 100 largest diameter trees per hectare.”

3.8 Line 87: Why 25 trees were selected? This statement needs to be supported or by references or by some kind of statistical inverstigation

Attended, line 131-134.

The text was rewritten.

3.9 Line 82: Beyond the reference, it would be informative, if you gave the equation of the Schöepfer function

Attended, lines 146-149.

The text was rewritten.

The Schöepfer model [40], characterized by the unsegmented 5th degree polynomial, was adjusted for each of the trees used in the study of FCP in the stem. That is, 148 trees (Table 2). It is not possible to bring all equations into the manuscript, but we can provide an appendix if the reviewer finds it pertinent.

3.10 Line 121: What do you mean by “diameter (dij) variable was isolated”? Please, explain.

Attended, line 159.

“…were fitted with the dependent variable diameter measured along the stem (dij), as proposed by Môra et al. [48].”

3.11 Lines 128 – 130: You have to support your statement by references.

Attended, lines 168-169.

3.12 Statistical evaluation of the models: You have used non-linear regression analysis, in order to estimate the coefficient values of the models. In the preprocessing phase, do you test any of the prerequisites of the regression analysis? If you have made the testing, you have to describe your results.

In the preprocessing phase, the prerequisites were tested for normality of the residuals by the Kolmogorov-Smirnov (KS) test statistic, independence of the residuals by the Durbin-Watson (DW) test statistic, and homogeneity of variances by the White (W) test statistic.

None of the equations met the prerequisites:

DW: p < 0.0001

KS: p < 0.01

W: p < 0.0001

However, considering the nature of the data, this result is expected, and some literature brings this about, as justified by the authors in lines 221-224: "Since the database contained repeated measurements in each tree - that is, hierarchical data - within the tree, the variance is heterogeneous and the correlation among the residues violates the basic assumptions (Özçelik et al. 2011)."

Rojo et al. (2005) brings that “The Kozak’s (2004) variable-exponent model fit presented residual autocorrelation due to its consecutive measurements in each tree. It is reasonable to expect that observations of each tree are spatially correlated, and that the assumption of independence of residuals is violated.”

Would the reviewer like the authors to bring these results to the item "2.5. Statistical evaluation of the models"?   

Özçelik, R.; Brooks, J.R.; Jiang, L. Modeling stem profile of Lebanon cedar, Brutian pine and Cilicica fir in Southern Turkey using nonlinear mixed-effects models. Eur. J. of For. Res. 2011, 130, 613-621. doi: 10.1007/s10342-010-0453-5.

Rojo A, Perales X, Sánchez-Rodríguez F, González-Alvarez JG & Von Gadow K. Stem taper functions for maritime pine (Pinus pinaster Ait.) in Galicia (Nortwestern Spain). E. J. For. Res. 2005, 124, 177-186. doi: 10.1007/s10342-005-0066-6.

Results and discussion

3.13 Lines 190 – 193: The statistical evaluation criteria of the fitted models, have to presented in the Materials and Methods section. 

Attended, lines 148-149.

The text was rewritten.

Conclusions: According to the length of the paper, this section is too small. I am sure that more conclusions derived from this study.

Attended.

The conclusion was rewritten.
